# FreeAdapt: Unleashing Diffusion Priors for Ultra-High-Definition Image Restoration

**Xiaoan Liu**[1], **Xinyi Liu**[1,2,3,*], **Yongjun Zhang**[1,2,3], **Yi Wan**[1,2,3], **Tingyun Li**[1], **Dongdong Yue**[1]

[1] School of Remote Sensing and Information Engineering, Wuhan University
[2] Technology Innovation Center for Collaborative Applications of Natural Resources Data in GBA
[3] Hubei LuoJia Laboratory
{xiaoanliu, liuxy0319, zhangyj, yi.wan, tingyunli, yueyisui}@whu.edu.cn

## Abstract

Latent Diffusion Models (LDMs) have recently shown great potential for image restoration owing to their powerful generative priors. However, directly applying them to ultra-high-definition image restoration (UHD-IR) often results in severe global inconsistencies and loss of fine-grained details, primarily caused by patch-based inference and the information bottleneck of the VAE. To overcome these issues, we present FreeAdapt, a plug-and-play framework that unleashes the capability of diffusion priors for UHD-IR. The core of FreeAdapt is a training-free Frequency Feature Synergistic Guidance (FFSG) mechanism, which introduces guidance at each denoising step during inference time. It consists of two modules: 1) Frequency Guidance (FreqG) selectively fuses phase information from a reference image in the frequency domain to enforce structural consistency across the entire image; 2) Feature Guidance (FeatG) injects global contextual information into the self-attention layers of the U-Net, effectively suppressing unrealistic textures in smooth regions and preserving local detail fidelity. In addition, FreeAdapt includes an optional VAE fine-tuning module, where skip connection further enhances the reconstruction of fine-grained textures. Extensive experiments demonstrate that our method achieves superior quantitative performance and visual quality compared to state-of-the-art UHD-IR approaches, and consistently delivers strong gains across multiple LDM-based backbones.

## 1 Introduction

With the rapid advancement of 4K/8K display and imaging technologies, the demand for Ultra-High-Definition (UHD) images is increasing dramatically (Wang et al., 2025; Li et al., 2023b; Zheng et al., 2021; Zhao et al., 2025; Liu et al., 2025b). However, in real-world capture, UHD images inevitably suffer from degradations such as low light, haze, blur, and noise, which are often caused by insufficient illumination, adverse weather conditions, or equipment limitations (Wang et al., 2025; 2024a). As a result, UHD image restoration (UHD-IR) has become a crucial yet highly challenging research field in computer vision, characterized by its massive resolution scale and requirements for preserving fine-grained details (Wang et al., 2025; Yu et al., 2024b).

To address the challenges of UHD-IR, researchers have proposed a variety of solutions (Zhao et al., 2025; Wang et al., 2024a; Liu et al., 2025b; Su et al., 2024; Liu et al., 2025a; Wu et al., 2024a). Existing studies primarily enhance restoration performance by designing innovative network architectures and developing advanced training paradigms. For instance, UHDformer (Wang et al., 2024a) employs a dual-path module to balance efficiency and accuracy, while ERR (Zhao et al., 2025) decomposes the restoration process into multiple stages for refined modeling. Although these methods have achieved remarkable performance, they unavoidably encounter bottlenecks, as merely modifying network structures is insufficient to overcome the inherently ill-posed nature of image restoration (Xu et al., 2024). For UHD-IR, how to leverage powerful diffusion priors to overcome the bottlenecks remains insufficiently explored.

Recently, Latent Diffusion Models (LDMs) (Rombach et al., 2022) have shown remarkable potential in low-level vision tasks owing to their powerful generative priors (Lin et al., 2024a; Wu et al., 2024c;b; Yue et al., 2025; Chen et al., 2025a; Sun et al., 2025). However, directly applying these pre-trained models to UHD-IR faces several technical bottlenecks. First, due to the high computa-

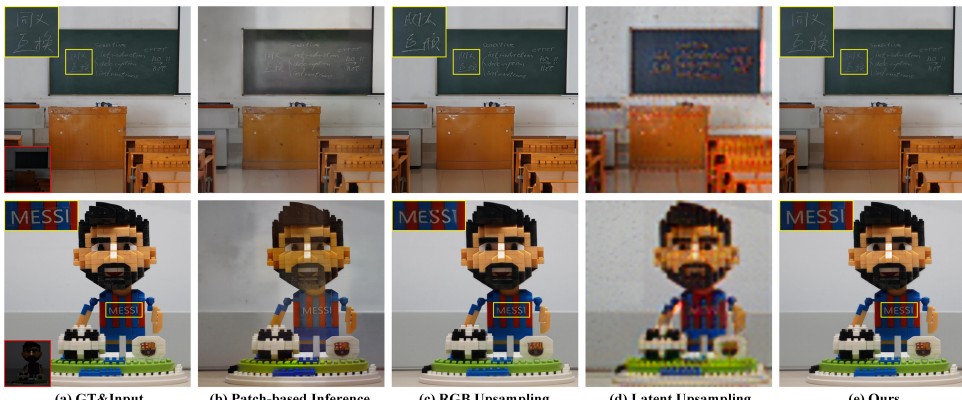

| (a) GT&Input | (b) Patch-based Inference | (c) RGB Upsampling | (d) Latent Upsampling | (e) Ours |

Figure 1: Visual comparison of different LDM-based strategies on the UHD-LL (Li et al., 2023a).

tional cost and memory demand of self-attention mechanisms, models cannot process an entire UHD image in a single pass, making patch-based inference unavoidable. As illustrated in Figure 1(b–d), this strategy often introduces stripe-like artifacts and color inconsistencies, while naive upsampling schemes typically cause blurring or structural distortions. Second, the absence of global context in patch-based inference amplifies the stochasticity of diffusion models, leading to inconsistent high-frequency details in textureless regions. Finally, the Variational Autoencoder (VAE) (Kingma & Welling, 2013), as a core component of LDMs, suffers from lossy compression that discards high-frequency information and thereby limits restoration fidelity.

To enable pre-trained LDMs to overcome these challenges, we introduce **FreeAdapt**, a unified framework that combines a plug-and-play, training-free guidance mechanism with an optional VAE fine-tuning (VAE-FT) module. FreeAdapt provides a cost-efficient way to unleash the potential of diffusion priors and adapt pre-trained LDMs and their extensions (e.g., ControlNet (Zhang et al., 2023)) to UHD-IR. The core of our approach is the Frequency Feature Synergistic Guidance (FFSG) mechanism, which enforces both global consistency and local detail fidelity at each step of patch-based denoising during inference time. Specifically, FFSG is composed of two complementary modules: 1) **Frequency Guidance (FreqG)**, which selectively fuses phase information from a low-resolution reference image in the frequency domain to ensure global structural consistency across patches; 2) **Feature Guidance (FeatG)**, which incorporates global context into the U-Net self-attention layers to constrain local detail generation and suppress high-frequency hallucinations. In addition, the optional **VAE-FT** module fine-tunes the VAE decoder with skip connection to ease the information bottleneck and improve the reconstruction of fine textures.

Our main contributions are summarized as follows:

- To the best of our knowledge, FreeAdapt is the first plug-and-play diffusion prior framework for the UHD-IR task, providing an effective and generalizable solution for pre-trained LDMs and their extensions.
- We propose a training-free, plug-and-play synergistic guidance mechanism that, through innovative frequency and feature guidance modules, effectively resolves the artifact and detail hallucination issues in UHD-IR, significantly improving both global consistency and local fidelity.
- By introducing skip connection and fine-tuning the VAE decoder, we successfully mitigate the VAE's information bottleneck and improve the fidelity of reconstructed details.
- Through extensive experiments across three representative LDM-based backbones (LDM (Rombach et al., 2022), StableSR (Wang et al., 2024b), and DiffBIR (Lin et al., 2024a)), we show that FreeAdapt consistently delivers strong performance improvements, achieving PSNR gains typically above 2 dB over patch-based inference. These gains stem from its ability to correct cross-patch inconsistencies and more effectively exploit pre-trained diffusion priors.

## 2 RELATED WORK

**Ultra-High-Definition Image Restoration**. UHD-IR has gained increasing attention due to the rapidly growing demand for high-resolution images (Li et al., 2023b; Liu et al., 2025b; Wang et al.,

2025; Zhao et al., 2025). UHDformer (Wang et al., 2024a) achieved a balance between performance and efficiency by introducing a dual-path architecture with correlation matching and channel modulation. DreamUHD (Liu et al., 2025b) employed a frequency-enhanced VAE, integrating Fourier and wavelet modules to improve detail fidelity. From a spectral perspective, ERR (Zhao et al., 2025) deconstructed the restoration process into three progressive stages, incorporating multiple structures for phased modeling. Although these methods achieved notable progress, relying on architectural innovations and training paradigms alone cannot fundamentally resolve the inherently ill-posed nature of image restoration, leading to inevitable performance bottlenecks (Xu et al., 2024). In the context of UHD-IR, the potential of leveraging diffusion priors has been overlooked. Therefore, this study aims to investigate and utilize the powerful generative priors in pre-trained models to enhance restoration performance for UHD-IR, specifically addressing the persistent issues of insufficient priors and the loss of fine-grained details.

**High-resolution Adaptation of Diffusion Models**. With the rapid advancement of diffusion models in image generation, high-resolution image synthesis and upscaling have become significant research hotspots (Tragakis et al., 2024; Huang et al., 2024). Existing approaches fall into two categories. The first (Ren et al., 2024; Zhang et al., 2025) retrains or fine-tunes models on high-resolution datasets, which requires substantial data and computational resources. The second (Bar-Tal et al., 2023; Du et al., 2024; Lin et al., 2024b; Huang et al., 2024; Zhang et al., 2024c; Qiu et al., 2024; Zhang et al., 2024b) follows a training-free paradigm that improves effective resolution by optimizing the inference procedure. Representative methods such as MultiDiffusion (Bar-Tal et al., 2023) generate large images by fusing multiple diffusion trajectories, while DemoFusion (Du et al., 2024) enhances visual coherence through progressive sampling and skip-residual refinement. AccDiffusion (Lin et al., 2024b) further emphasizes semantic alignment when guiding patch-level generation. These techniques are highly effective for high-resolution generation, where outputs only need to be perceptually plausible. In contrast, UHD restoration requires strict structural fidelity to the degraded input, and any hallucinated or altered content violates the restoration objective. As a result, generation-oriented strategies often struggle to maintain input–output consistency in UHD-IR. Meeting restoration requirements therefore relies on two principles: selective injection of reliable global information and preservation of structural alignment throughout denoising. Our frequency and feature guidance modules follow these principles, enabling diffusion priors to maintain global coherence while faithfully preserving input-consistent details.

**Diffusion Prior-Based Image Restoration**. In recent years, with the breakthrough development of LDMs (Rombach et al., 2022) in image and video generation, their capability as powerful generative priors has gradually been introduced into low-level vision tasks (Wang et al., 2024b; Lin et al., 2024a; Wu et al., 2024c; Yang et al., 2024; Yu et al., 2024a; Ai et al., 2024; Zhang et al., 2024a; Chen et al., 2025b; Arora et al., 2025). StableSR (Wang et al., 2024b) fine-tuned the pre-trained model with a time-aware encoder and feature modulation mechanism to achieve high-quality image super-resolution. DiffBIR (Lin et al., 2024a) adopted a two-stage strategy to extend the adaptability of diffusion models to blind image restoration tasks. SeeSR (Wu et al., 2024c) designed a degradation-aware text prompt generator to guide more refined super-resolution reconstruction. SUPIR (Yu et al., 2024a) incorporated prompts generated by multimodal large language models and employed a degradation-robust adapter for prior control. Although these methods have achieved notable advances in image restoration, they primarily concentrated on optimizing performance within native resolution. For UHD-IR, directly applying existing models often results in artifacts such as distortions, color inconsistencies, and the loss of fine details caused by the VAE. Therefore, designing a plug-and-play, artifact-free, and universal image restoration approach for UHD images, remains an important research direction.

## 3 METHODOLOGY

**Preliminary**. LDMs (Rombach et al., 2022) are text-to-image diffusion models that perform denoising in a latent space. Specifically, LDMs employ a pre-trained VAE (Kingma & Welling, 2013) to encode an image into a latent representation $z_0$, followed by training a denoising U-Net $\epsilon_\theta$ in the latent space. The training objective of the LDM is formulated as:

$$\mathcal{L}_{ldm} = \mathbb{E}_{z,c,t,\epsilon}[||\epsilon - \epsilon_\theta(\sqrt{\bar{\alpha}_t}z + \sqrt{1 - \bar{\alpha}_t}\epsilon, c, t)||_2^2] \tag{1}$$

where $\epsilon \sim \mathcal{N}(0, \mathbf{I})$ denotes the ground-truth noise at timestep $t$, $c$ represents the conditional information, and $\bar{\alpha}_t$ is the diffusion coefficient defined in DDPM (Ho et al., 2020).

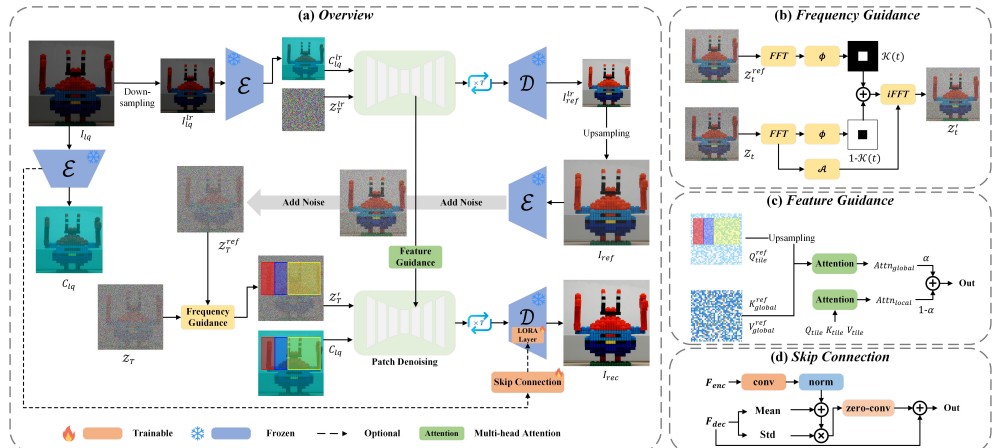

Figure 2: Overview of the proposed FreeAdapt framework. The degraded UHD input $I_{lq}$ is downsampled and passed through a pre-trained LDM to obtain a reference latent $z_0^{ref}$ for global structural guidance. During iterative patch-based denoising of $z_t$, frequency guidance (FreqG) fuses the phase spectrum of $z_t$ and $z_t^{ref}$ to enforce cross-patch consistency, while feature guidance (FeatG) injects global context into U-Net attention layers to suppress artifacts. Finally, the latent $z_t'$ is decoded by $D$, where an optional VAE fine-tuning (VAE-FT) module with skip connection enhances high-frequency details, producing the restored UHD output $I_{rec}$.

**Objective**. To address the challenges of adapting pre-trained LDMs for UHD-IR, we introduce a novel plug-and-play FreeAdapt framework. Our primary objective is to resolve the prevalent issues of artifacts, global inconsistencies and detail loss that arise when directly applying pre-trained LDMs to UHD-IR, without modifying or fine-tuning of the denoising U-Net.

**Overview**. As shown in Figure 2, the core of FreeAdapt is a training-free guidance mechanism that operates during the iterative, patch-based denoising process. This mechanism integrates FreqG and FeatG modules to enforce global consistency and preserve local detail fidelity. Additionally, to overcome the inherent high-frequency information loss of the VAE, we introduce an optional VAE-FT module that fine-tunes the VAE decoder with lightweight modifications to further boost the fidelity of the restored images.

## 3.1 FREQUENCY FEATURE SYNERGISTIC GUIDANCE

FFSG is a training-free, plug-and-play mechanism compatible with pre-trained LDMs and their extensions, such as ControlNet (Zhang et al., 2023). It is designed to address the global inconsistencies and local detail distortions that arise from patch-based inference. As illustrated in Figure 2, within each denoising step, we employ frequency-domain constraints to stabilize low-frequency structures and textures, while a feature-level attention module suppresses high-frequency artifacts and promotes cross-patch consistency. The overall mechanism consists of two main stages: **Reference Image Generation** and **Guided High-Resolution Iterative Denoising**.

**Reference Image Generation**. LDMs are typically trained at a fixed resolution, making their direct application to UHD images prone to structural distortions and artifacts. To address this, we first generate a reference image that provides reliable low-frequency structural information by leveraging the inherent image-to-image capability of the diffusion backbone. Specifically, the degraded UHD input $I_{lq}$ is downsampled to the native training resolution (e.g., 512×512), encoded by the VAE to obtain the conditioning latent, and then used to guide a single standard denoising process of the pre-trained LDM, producing a clean latent representation $z_0^{lr}$ with coherent content and structure. This latent is decoded into the pixel domain, upsampled back to the UHD resolution to form $I_{ref}$, and re-encoded by the VAE encoder to obtain $z_0^{ref}$ for global guidance. Compared with cascaded multi-resolution approaches (Du et al., 2024; Lin et al., 2024b), our strategy provides a structurally reliable reference using only a single denoising process.

**Guided High-Resolution Iterative Denoising**. To address GPU memory limitations in UHD-IR, we adopt a patch-based denoising strategy. At each denoising step $t$, multiple small patches are cropped from the current high-resolution latent representation and denoised individually. Overlap-

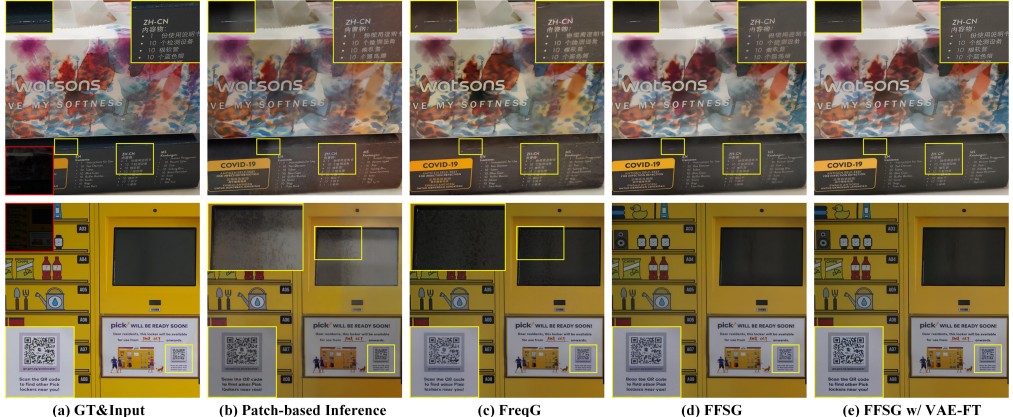

| (a) GT&Input | (b) Patch-based Inference | (c) FreqG | (d) FFSG | (e) FFSG w/ VAE-FT |

Figure 3: Visual comparison and ablation study of FreeAdapt on the UHD-LL. (b) Patch-based inference suffers from color inconsistency; (c) FreqG contains high-frequency noise; (d) FFSG still exhibits detail loss.

ping regions are blended by smooth averaging to maintain consistency across patch boundaries. Within each denoising step, **FreqG** and **FeatG** are integrated into the denoising process, jointly improving both global structural coherence and local detail fidelity.

**Frequency Guidance**. To overcome the structural inconsistencies introduced by patch-based denoising, as shown in Figure 3(b), we incorporate FreqG into the iterative denoising process. At each step $t$, both the current latent representation $z_t$ and the noised reference latent $z_t^{ref}$ are transformed into the frequency domain using Fast Fourier Transformation (FFT), yielding their respective amplitude and phase spectrum:

$$\mathcal{FFT}(z_t) = \mathcal{A}_t e^{i\phi_t} \tag{2}$$

$$\mathcal{FFT}(z_t^{ref}) = \mathcal{A}_t^{ref} e^{i\phi_t^{ref}} \tag{3}$$

To ensure that the global structure of the reference is effectively preserved without interfering with texture generation, only the phase components are fused. Specifically, a dynamically low-pass filter $\mathcal{K}(t)$ is applied to weight the two phase spectrum:

$$\overline{\phi}_t = \arctan\left((1 - \mathcal{K}(t))e^{i\phi_t} + \mathcal{K}(t)e^{i\phi_t^{ref}}\right) \tag{4}$$

$$\mathcal{K}(t) = \begin{cases} \frac{t}{T}, & \text{if } |x - \frac{w}{2}| < w \cdot c \cdot \frac{t}{T} \text{ and } |y - \frac{h}{2}| < h \cdot c \cdot \frac{t}{T} \\ 0, & \text{otherwise} \end{cases} \tag{5}$$

where $c$ is a hyperparameter (default 0.15). The filter $\mathcal{K}(t)$, defined in Eq. (5), gradually decreases as the denoising step progresses, adaptively balancing global structural constraints with flexibility for detail generation. The corrected latent is then reconstructed by combining the original amplitude spectrum $\mathcal{A}_t$ with the fused phase spectrum $\overline{\phi}_t$ through inverse FFT:

$$z_t' = i\mathcal{FFT}(\mathcal{A}_t e^{i\overline{\phi}_t}) \tag{6}$$

**Feature Guidance**. While FreqG enforces global low-frequency consistency, it cannot constrain high-frequency details generated independently within each patch. As shown in Figure 3(c), in textureless regions this randomness often introduces spurious details, leading to visual noise or artifacts. To address this issue, we introduce FeatG module that injects global contextual information into the self-attention layers of the U-Net. This allows each patch to reference the global semantics provided by the guidance image, thereby promoting inter-patch coherence and suppressing unrealistic artifacts. Specifically, we first compute the Query ($Q_{tile}$), Key ($K_{tile}$), and Value ($V_{tile}$) of the current high-resolution patch to obtain local attention:

$$Attn_{local} = softmax\left(\frac{Q_{tile} \cdot K_{tile}^T}{\sqrt{d}}\right) V_{tile} \tag{7}$$

where $d$ is the feature dimension. In parallel, we extract the patch-aligned query $Q_{tile}^{ref}$ from the reference, together with global keys $K_{global}^{ref}$ and values $V_{global}^{ref}$, and compute the global attention:

$$Attn_{global} = softmax\left(\frac{\mathcal{U}(Q_{tile}^{ref}) \cdot K_{global}^{ref}{}^{T}}{\sqrt{d}}\right) V_{global}^{ref} \qquad (8)$$

where $\mathcal{U}$ denotes an upsampling operation. The final output is obtained by linearly blending the two attentions:

$$Attn_{final} = (1 - \alpha) \cdot Attn_{local} + \alpha \cdot Attn_{global} \qquad (9)$$

where $\alpha$ is set to 0.2 by default. Importantly, this operation is applied to the 3rd–8th decoder layers of the U-Net.

## 3.2 VAE FINE-TUNING

In LDMs, VAE is responsible for perceptual compression, but the lossy compression characteristic causes the loss of high-frequency details such as fine textures and text at high resolutions as illustrated in Figure 3(d). This limitation makes the VAE a major bottleneck for UHD-IR. To address this issue, we introduce an optional VAE-FT module that strengthens the decoder's ability to recover fine details while keeping both the encoder and the U-Net frozen, ensuring that the diffusion process remains fully training-free.

During fine-tuning, both a low-quality image and its high-quality counterpart are passed through the shared VAE encoder, producing a high-quality latent representation along with residual features extracted from the degraded input. The decoder receives the high-quality latent together with these residual features through skip connections, which provide structural cues that help restore information lost during encoding. Through this training strategy, VAE-FT learns a task-level prior for detail reconstruction without learning the restoration task itself. Because this prior is task specific rather than model specific, the fine-tuned decoder can be applied across different diffusion backbones without further training, serving as a lightweight auxiliary component that complements rather than alters the diffusion prior. With this training framework in place, we implement VAE-FT using an enhanced decoder designed to efficiently integrate these residual cues.

As shown in Figure 2, we enhance the VAE decoder by introducing skip connection combined with parameter-efficient LoRA (Hu et al., 2022). Encoder features from the degraded input are first refined through adaptive instance normalization (AdaIN) (Huang & Belongie, 2017), which suppresses degradation while retaining structural details, and are then injected into the corresponding upsampling layers of the decoder via Zero-Convolution modules (Zhang et al., 2023). The fine-tuning is supervised by a composite loss:

$$L = L_{dwt} + L_{lpips} + L_{ssim} + L_{gan} \qquad (10)$$

where $L_{dwt}$ is an L2 loss in the Discrete Wavelet Transform domain for reconstructing high-frequency details, $L_{lpips}$ ensures perceptual similarity, $L_{ssim}$ maintains structural consistency, and $L_{gan}$ introduces adversarial feedback to improve realism and sharpness.

## 4 EXPERIMENTS

### 4.1 EXPERIMENTAL SETUP

**Datasets**. To comprehensively evaluate the performance of our proposed method on UHD-IR tasks, we conduct experiments on multiple publicly available benchmark datasets. For low-light enhancement, we use the UHD-LL dataset (Li et al., 2023a). For image dehazing, we evaluate performance on the UHD-Haze dataset (Wang et al., 2024a). For image deblurring, we adopt the UHD-Blur dataset (Wang et al., 2024a).

**Evaluation Metrics**. We employ a combination of reference-based and no-reference metrics to provide a comprehensive evaluation of restoration results. PSNR and SSIM (Wang et al., 2004) are used as conventional measures, assessing pixel-level reconstruction accuracy and structural similarity, respectively. LPIPS (Zhang et al., 2018) and DISTS (Ding et al., 2020) are adopted as perceptual metrics to better reflect human visual judgments of image quality. In addition, to further evaluate perceptual quality in a reference-free setting, we use no-reference image quality assessors including CLIPIQA (Wang et al., 2023), MUSIQ (Ke et al., 2021), and MANIQA (Yang et al., 2022).

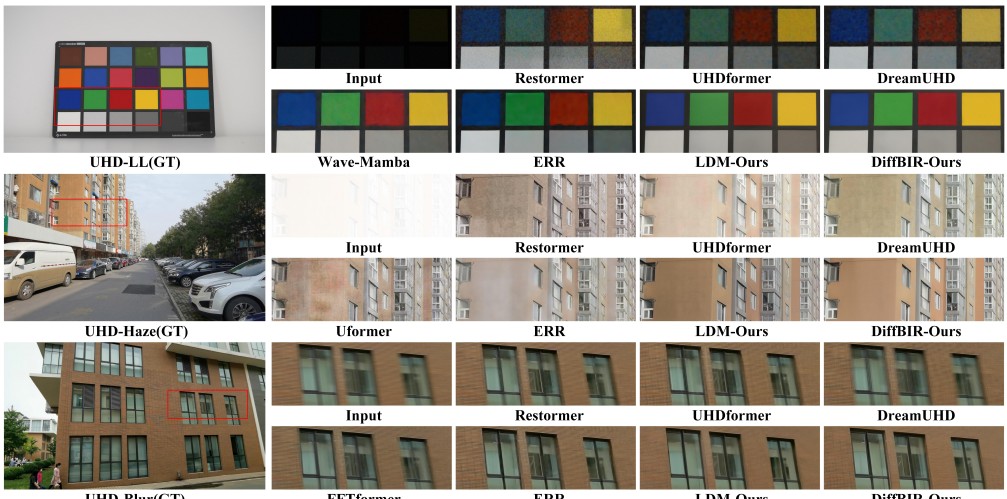

Figure 4: Visual comparison of the proposed FreeAdapt against state-of-the-art approaches.

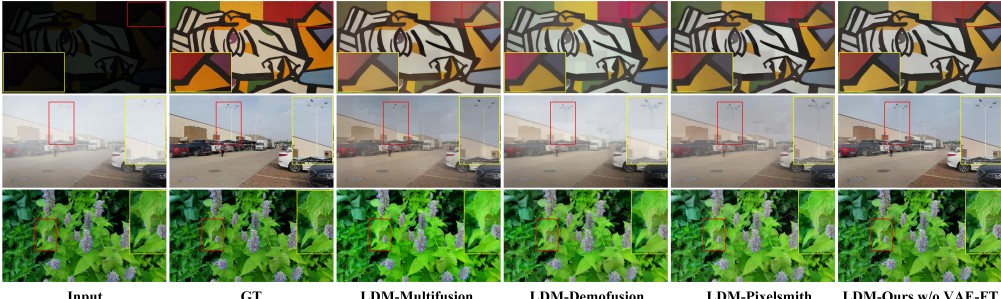

Figure 5: Visual comparison of training-free diffusion model adaptation methods based on LDM on three UHD-IR datasets: UHD-LL (first row), UHD-Haze (second row), and UHD-Blur (third row).

## 4.2 EXPERIMENTAL RESULTS

We conduct extensive quantitative and qualitative experiments on multiple UHD-IR tasks to demonstrate the effectiveness of diffusion priors and that our method more effectively leverages pre-trained diffusion models in UHD scenarios. For a fair and comprehensive evaluation, we compare our approach against two categories of methods: (1) task-specific restoration methods, including representative approaches that achieve leading performance on individual restoration tasks, such as SwinIR (Liang et al., 2021), Restormer (Zamir et al., 2022), Uformer (Wang et al., 2022), UHDFormer (Wang et al., 2024a), UHDFour (Li et al., 2023a), Wave-Mamba (Zou et al., 2024), FFTformer (Kong et al., 2023), 4KDehazing (Xiao et al., 2024), DreamUHD (Liu et al., 2025b), and ERR (Zhao et al., 2025); (2) training-free diffusion adaptation methods that adapt standard diffusion models to high-resolution images without additional training, such as MultiDiffusion (Bar-Tal et al., 2023), DemoFusion (Du et al., 2024), and PixelSmith (Tragakis et al., 2024). Moreover, to further validate the generality of FreeAdapt under different diffusion architectures, we conduct experiments using three representative diffusion backbones, including a classical latent diffusion model (LDM) (Rombach et al., 2022), the restoration-oriented DiffBIR (Lin et al., 2024a), and the super-resolution oriented StableSR (Wang et al., 2024b), and evaluate them under the same UHD-IR setting.

**Low-Light Image Enhancement**. As shown in Table 1, our method outperforms advanced approaches on no-reference metrics and the perceptual metrics DISTS. Although it yields lower scores than non-diffusion methods on full-reference metrics, this is largely because those methods are trained end-to-end with L2 or perceptual losses that are directly aligned with these metrics. However, such optimization often sacrifices the realism of generated details, producing overly smoothed results (Yang et al., 2024; Wang et al., 2024b; Wu et al., 2024c). The visual comparisons in Figure 4 clearly demonstrate that our diffusion prior–based methods restore richer and more realistic details, whereas competing methods generally produce blurred outputs, which contradicts the core objective of UHD-IR: recovering fine-grained details.

Table 1: Quantitative comparison of the proposed method against various state-of-the-art methods. The symbols ↑ and ↓ respectively represent that higher or lower values indicate better performance. Bold represents the best and underline represents the second best.

| Dataset | Method | PSNR ↑ | SSIM ↑ | LPIPS ↓ | DISTS ↓ | CLIPIQA ↑ | MUSIQ ↑ | MANIQA ↑ |
|---|---|---|---|---|---|---|---|---|
| UHD-LL | Uformer | 22.57 | 0.904 | 0.258 | 0.170 | 0.335 | 32.63 | 0.259 |
| | Restormer | 23.46 | 0.906 | 0.268 | 0.160 | 0.402 | 36.05 | 0.265 |
| | Swinir | 22.12 | 0.905 | 0.214 | 0.114 | 0.385 | 33.51 | 0.305 |
| | UHDFour | 28.59 | 0.918 | 0.235 | 0.140 | 0.411 | 28.08 | 0.280 |
| | UHDformer | 27.10 | 0.926 | 0.232 | 0.138 | 0.360 | 35.83 | 0.304 |
| | Wave-Mamba | 29.84 | 0.941 | 0.185 | 0.117 | 0.410 | 41.78 | 0.337 |
| | DreamUHD | 27.73 | 0.929 | 0.220 | 0.132 | 0.378 | 38.52 | 0.311 |
| | ERR | 27.57 | 0.933 | 0.214 | 0.148 | 0.501 | 42.28 | 0.344 |
| | LDM-Ours | 22.21 | 0.887 | 0.253 | 0.101 | 0.569 | 49.07 | 0.372 |
| | StableSR-Ours | 22.42 | 0.887 | 0.244 | 0.093 | 0.560 | 48.35 | 0.364 |
| | DiffBIR-Ours | 23.99 | 0.900 | 0.233 | 0.092 | 0.564 | 48.37 | 0.364 |
| UHD-Haze | Uformer | 23.38 | 0.937 | 0.136 | 0.069 | 0.283 | 31.97 | 0.267 |
| | Restormer | 23.10 | 0.930 | 0.157 | 0.076 | 0.294 | 33.69 | 0.264 |
| | Swinir | 24.09 | 0.943 | 0.101 | 0.038 | 0.288 | 32.07 | 0.285 |
| | UHDformer | 22.58 | 0.942 | 0.118 | 0.049 | 0.301 | 31.72 | 0.288 |
| | 4KDehazing | 22.50 | 0.906 | 0.185 | 0.145 | 0.329 | 35.60 | 0.281 |
| | DreamUHD | 24.36 | 0.945 | 0.116 | 0.048 | 0.282 | 33.08 | 0.280 |
| | ERR | 25.10 | 0.949 | 0.119 | 0.051 | 0.282 | 31.17 | 0.292 |
| | LDM-Ours | 21.59 | 0.934 | 0.104 | 0.044 | 0.403 | 44.38 | 0.343 |
| | StableSR-Ours | 22.80 | 0.945 | 0.092 | 0.033 | 0.393 | 43.63 | 0.333 |
| | DiffBIR-Ours | 25.50 | 0.953 | 0.077 | 0.028 | 0.404 | 42.18 | 0.337 |
| UHD-Blur | Uformer | 28.88 | 0.851 | 0.205 | 0.103 | 0.284 | 29.21 | 0.250 |
| | Restormer | 29.57 | 0.860 | 0.210 | 0.115 | 0.300 | 28.86 | 0.243 |
| | Swinir | 28.30 | 0.834 | 0.190 | 0.086 | 0.269 | 28.10 | 0.236 |
| | FFTformer | 26.28 | 0.825 | 0.215 | 0.105 | 0.288 | 31.22 | 0.236 |
| | UHDformer | 28.81 | 0.843 | 0.233 | 0.127 | 0.299 | 27.31 | 0.233 |
| | DreamUHD | 27.15 | 0.808 | 0.284 | 0.172 | 0.265 | 24.33 | 0.205 |
| | ERR | 29.71 | 0.861 | 0.206 | 0.106 | 0.267 | 29.28 | 0.251 |
| | LDM-Ours | 26.91 | 0.828 | 0.166 | 0.076 | 0.347 | 34.39 | 0.278 |
| | StableSR-Ours | 27.42 | 0.831 | 0.162 | 0.077 | 0.364 | 36.10 | 0.278 |
| | DiffBIR-Ours | 28.16 | 0.851 | 0.145 | 0.059 | 0.378 | 38.64 | 0.290 |

**Image Dehazing**. In the evaluation on the UHD-Haze dataset, the DiffBIR-Ours model demonstrates comprehensive superiority. As shown in Table 1, it achieves the best performance on both reference-based metrics and perceptual metrics. Notably, on the perceptual metric DISTS, our approach delivers a remarkable improvement of about 26.3% over the second-best method SwinIR. Furthermore, the visual results in Figure 4 illustrate that our method more effectively restores color saturation and contrast while avoiding distortions and detail loss, providing strong evidence for the effectiveness of diffusion priors.

**Image Deblurring**. As shown in Table 1, our adapted diffusion prior-based model also achieves strong performance. On the perceptual metric LPIPS, DiffBIR-Ours achieves an improvement of about 29.6% over ERR, highlighting the effectiveness of diffusion priors. Moreover, the visual comparisons in Figure 4 demonstrate that our method can effectively handle complex motion blur, producing images with sharp edges and well-preserved textures. In contrast, competing methods often suffer from ringing artifacts or fail to completely remove blur.

**Comparison with Training-Free Diffusion Adaptation Methods**. To further evaluate the generality of our guidance mechanism, Table 2 compares training-free diffusion adaptation strategies across three representative backbones: LDM, DiffBIR, and StableSR. Across all UHD-IR tasks, our FFSG modules consistently yield clear improvements over patch-based inference, as reflected in the *FFSG Gain* rows, and outperform MultiDiffusion, DemoFusion, and PixelSmith on PSNR, LPIPS, and MUSIQ. A key factor behind this performance gap is that existing methods were originally designed for high-resolution generation, where perceptual plausibility is prioritized, whereas UHD restoration demands strict consistency with the degraded input. Our FFSG mechanism is explicitly tailored for restoration, enforcing global structural coherence and input-aligned texture synthesis.

Table 2: Quantitative comparison of diffusion model adaptation methods using LDM (Rombach et al., 2022), StableSR (Wang et al., 2024b), and DiffBIR (Lin et al., 2024a). PI indicates patch-based inference. Bold and underlined entries denote the best and second-best results, respectively. Red values represent positive gains, while blue values indicate negative gains.

| Model | UHD-LL | UHD-Haze | UHD-Blur |
|---|---|---|---|
| | PSNR / LPIPS / MUSIQ | PSNR / LPIPS / MUSIQ | PSNR / LPIPS / MUSIQ |
| LDM-PI | 18.91 / 0.386 / 44.89 | 19.27 / 0.190 / 42.63 | 23.61 / 0.213 / 35.37 |
| LDM-Multidiffusion | 20.13 / 0.399 / 32.41 | 18.59 / 0.240 / 38.71 | 25.16 / 0.364 / 26.49 |
| LDM-Demofusion | 21.74 / 0.417 / 23.09 | 19.40 / 0.292 / 32.17 | 23.81 / 0.400 / 22.27 |
| LDM-Pixelsmith | 20.64 / 0.397 / 31.15 | 20.64 / 0.224 / 42.97 | 24.07 / 0.323 / 29.75 |
| LDM-Ours w/o VAE-FT | 21.88 / 0.283 / 45.67 | 21.37 / 0.163 / **44.75** | 26.58 / 0.198 / **35.52** |
| LDM-Ours | **22.21** / **0.253** / **49.07** | **21.59** / **0.104** / 44.38 | **26.91** / **0.166** / 34.39 |
| FFSG Gain | +2.96 / -0.103 / +0.78 | +2.11 / -0.028 / +2.13 | +2.97 / -0.015 / +0.15 |
| VAE-FT Gain | +0.33 / -0.030 / +3.40 | +0.22 / -0.059 / -0.38 | +0.33 / -0.032 / -1.13 |
| StableSR-PI | 19.14 / 0.369 / 45.34 | 19.85 / 0.182 / 40.28 | 24.10 / 0.202 / 35.74 |
| StableSR-Multidiffusion | 19.52 / 0.355 / 37.71 | 20.14 / 0.228 / 40.37 | 24.80 / 0.259 / 32.47 |
| StableSR-Demofusion | 21.59 / 0.393 / 25.65 | 20.17 / 0.211 / 34.25 | 25.31 / 0.382 / 23.38 |
| StableSR-Pixelsmith | 20.79 / 0.361 / 33.48 | 20.92 / 0.198 / 41.26 | 25.65 / 0.297 / 32.62 |
| StableSR-Ours w/o VAE-FT | 21.96 / 0.270 / 47.20 | 22.51 / 0.142 / **44.02** | 27.07 / 0.195 / **36.80** |
| StableSR-Ours | **22.42** / **0.244** / **48.35** | **22.80** / **0.092** / 43.63 | **27.42** / **0.162** / 36.10 |
| FFSG Gain | +2.82 / -0.098 / +1.86 | +2.65 / -0.040 / +3.74 | +2.97 / -0.007 / +1.06 |
| VAE-FT Gain | +0.46 / -0.026 / +1.14 | +0.29 / -0.051 / -0.40 | +0.35 / -0.033 / -0.70 |
| DiffBIR-PI | 21.61 / 0.280 / 45.38 | 23.35 / 0.143 / 41.58 | 26.65 / 0.174 / 38.09 |
| DiffBIR-Multidiffusion | 22.18 / 0.285 / 40.24 | 23.35 / 0.141 / 38.21 | 24.87 / 0.176 / 32.73 |
| DiffBIR-Demofusion | 23.22 / 0.335 / 24.82 | 23.44 / 0.224 / 32.59 | 26.06 / 0.280 / 26.74 |
| DiffBIR-Pixelsmith | 23.45 / 0.316 / 32.44 | 23.90 / 0.225 / 40.20 | 24.79 / 0.291 / 29.11 |
| DiffBIR-Ours w/o VAE-FT | 23.66 / 0.252 / 45.85 | 24.96 / 0.127 / **42.41** | 27.68 / 0.175 / **38.97** |
| DiffBIR-Ours | **23.99** / **0.233** / **48.37** | **25.50** / **0.077** / 42.18 | **28.16** / **0.145** / 38.64 |
| FFSG Gain | +2.05 / -0.028 / +0.48 | +1.61 / -0.017 / +0.82 | +1.03 / +0.001 / +0.87 |
| VAE-FT Gain | +0.33 / -0.019 / +2.52 | +0.54 / -0.050 / -0.22 | +0.48 / -0.030 / -0.33 |

As shown in Figure 5, generation-oriented baselines often introduce blurring, distortions, or inconsistent textures. In contrast, diffusion models equipped with FFSG produce sharper details, more stable structures, and visually coherent results that remain faithful to the input.

## 4.3 ABLATION STUDY

Table 3: Ablation study of the proposed methods.

| FreqG | FeatG | VAE-FT | PSNR ↑ | LPIPS ↓ |
|---|---|---|---|---|
| × | × | × | 18.91 | 0.386 |
| ✓ | × | × | 21.76 | 0.314 |
| ✓ | ✓ | × | 21.88 | 0.283 |
| ✓ | ✓ | ✓ | **22.21** | **0.253** |

Table 4: Comparison of Fusion Methods of LDM.

| Fusion method | SSIM ↑ | DISTS ↓ |
|---|---|---|
| Patch-based Inference | 0.823 | 0.145 |
| Skip Residual | 0.839 | 0.312 |
| FFT Fusion | 0.863 | 0.187 |
| FeaqG | **0.865** | **0.121** |

To validate the effectiveness of each proposed component, we conduct a series of comprehensive ablation experiments. All experiments are performed on the UHD-LL (Li et al., 2023a), with pre-trained LDM equipped with a standard patch-based denoising strategy serving as the baseline model.

**Effectiveness of Frequency Guidance**. As shown in Figure 1(b), Figure 3(b) and Figure 6, the baseline model produces severe stitching artifacts and color inconsistencies when directly applied with patch-based inference. After incorporating our proposed FreqG, global consistency is significantly improved, and Table 3 further shows notable improvements in PSNR and LPIPS. To validate the superiority of our fusion strategy, we also compare it with alternative designs, including spatial-domain fusion (skip residual from DemoFusion(Du et al., 2024)) and FFT spectrum fusion (Yang et al., 2025). As presented in Figure 6, the former fails to distinguish the roles of low- and high-frequency information during the diffusion process, resulting in blurred outputs, while the latter leads to severe color distortions. In addition, the quantitative results in Table 4 show that both alter-

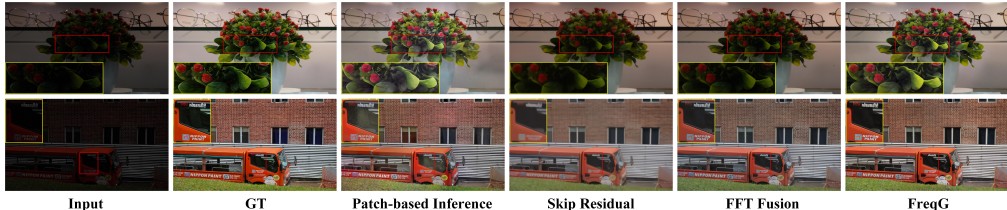

Figure 6: Qualitative comparison of different diffusion fusion methods.

natives substantially degrade perceptual quality, with much worse DISTS scores compared to ours. These comparisons clearly demonstrate the effectiveness of our phase-only fusion strategy.

**Effectiveness of Feature Guidance**. Building upon FreqG alone, we further incorporate the proposed FeatG module. As shown in Table 3, this addition yields further improvements on perceptual metrics such as LPIPS. The visual comparisons in Figure 3 also demonstrate that FeatG effectively suppresses hallucinated noise and unrealistic textures in smooth regions, producing more faithful and coherent local details.

**Effectiveness of VAE Decoder Fine-tuning**. Finally, we validate the effectiveness of the proposed VAE fine-tuning module. Within the full guidance mechanism, we compare the performance of the standard VAE with our fine-tuned VAE decoder. As shown in Table 2 and Table 3, the fine-tuned decoder achieves significant improvements on PSNR and LPIPS, while the visual results in Figure 3 further confirm its superiority in reconstructing fine details. Moreover, to examine the role of the introduced skip connection, we conduct additional ablation studies. The comparisons in Figure 7 clearly indicate that skip connection effectively alleviates the information bottleneck and enable the recovery of sharper high-frequency details, whereas removing them leads to noticeable detail loss.

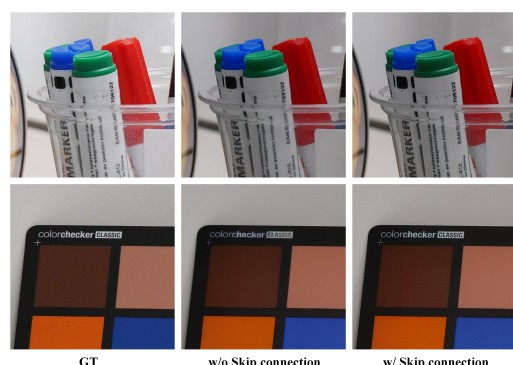

Figure 7: Ablation study on the skip connection in the VAE-FT module.

## 5 CONCLUSION

In this paper, we propose FreeAdapt, a plug-and-play framework designed to unleash the potential of pre-trained diffusion priors for UHD-IR. FreeAdapt integrates FreqG to correct global low-frequency structures and colors, ensuring cross-patch consistency, while FeatG introduces global context into the U-Net attention layers to suppress unrealistic high-frequency details in smooth regions. In addition, we design an optional VAE-FT module, where skip connection further enhances the reconstruction of fine textures. Extensive experiments demonstrate that our method not only achieves significant improvements in perceptual metrics over state-of-the-art restoration methods but also consistently outperforms other diffusion adaptation approaches, highlighting its superiority in fully exploiting diffusion priors.

**Limitations and Future Work**. Despite its effectiveness, FreeAdapt has limitations. As an iterative denoising approach, it is time-consuming and computationally heavy. In future work, we will distill our guidance into one- or few-step generative models for efficiency and extend its applicability beyond U-Net–based designs to emerging frameworks such as Diffusion Transformers (DiTs).

**Practical Applicability.** Although iterative diffusion sampling is computationally demanding, LDMs+FreeAdapt is well suited for offline, quality-oriented UHD restoration workflows where visual fidelity takes precedence over runtime. Representative scenarios include film and television remastering, digital cultural heritage restoration, and large-scale remote sensing analysis, all of which routinely operate under non–real-time constraints and can therefore benefit from the superior perceptual fidelity enabled by diffusion priors.

**Acknowledgment**. This work was partially supported by the National Natural Science Foundation of China (Grant No. 42530106, 42571507), the State Key Laboratory of Micro-Spacecraft Rapid Design and Intelligent Cluster (Grant No. MS01240125).

**Ethics Statement**. This work aims to enhance the visual quality of ultra-high-definition images, with primary applications in consumer photography, media content production. The objective is to mitigate quality degradation caused by hardware limitations or challenging environmental conditions, thereby contributing positively to society. Given the increasing versatility of image restoration technologies, we explicitly advocate against their misuse, such as fabricating misleading content or employing them for malicious purposes.

**Reproducibility Statement**. We state that FreeAdapt is highly reproducible. Complete implementation details are provided in Appendix A. All critical hyperparameters and algorithmic procedures necessary for reproducing our results are thoroughly documented in the Methods section and the supplementary appendix.

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

# A  IMPLEMENTATION DETAILS

We adopt Stable Diffusion 2.1-base as the pre-trained generative prior in our experiments. During both the fine-tuning of the VAE-FT module and the inference of all compared methods, UHD images are cropped into $512 \times 512$ patches with a stride of 256. For the VAE-FT module, we apply parameter-efficient LoRA to the VAE decoder with a rank of 16. Training is conducted on paired degraded-clean images from the UHD-LL (Li et al., 2023a), UHD-Haze (Wang et al., 2024a), and UHD-Blur (Wang et al., 2024a) datasets. The learning rate is set to $2 \times 10^{-5}$, with a batch size of 8, for a total of 60,000 steps. Notably, we train a single unified VAE-FT module for all restoration tasks. This module is directly applied across different tasks and even when switching base models (e.g., from LDM to DiffBIR), without requiring retraining, which demonstrates strong generalization ability. All experiments are implemented on four NVIDIA RTX 3090 GPUs.

For the backbone diffusion models, we employ task-specific pre-trained LDM (Rombach et al., 2022), StableSR (Wang et al., 2024b) and DiffBIR (Lin et al., 2024a) to validate the broad adaptability of the FreeAdapt framework. Across all experiments, the number of denoising steps $T$ is fixed to 50.

We provide the complete inference pipeline of our proposed **FreeAdapt** framework in Algorithm 1. The process begins with the generation of a global reference latent representation $z_0^{ref}$, which provides reliable low-frequency structural information. Subsequently, during the high-resolution iterative denoising loop, FreqG ($G_{\text{freq}}$) and FeatG ($G_{\text{feat}}$) modules are jointly integrated at each step to progressively correct global inconsistencies and suppress unrealistic high-frequency artifacts, leading to high-fidelity UHD restoration.

---

**Algorithm 1:** FreeAdapt Inference Pipeline

**Input:** Degraded UHD image $I_{lq}$, pre-trained LDM ($\epsilon_\theta$, VAE), total denoising steps $T$
**Output:** Restored UHD image $I_{rec}$
```
// Stage 1:  Reference Generation
```
$I_{lr} \leftarrow \text{Downsample}(I_{lq})$ ;                      // Downsample to native resolution
$c_{lr} \leftarrow \text{VAE}_{encode}(I_{lr})$ ;            // Encode the low-res image as condition
$z_T^{lr} \sim \mathcal{N}(0, I)$ ;          // Initialize Gaussian noise in latent space
**for** $t = T$ **to** 1 **do**
$\quad\lfloor\ z_{t-1}^{lr} \leftarrow \epsilon_\theta(z_t^{lr}, t, c_{lr})$ ;      // Iterative denoising with LDM condition
$I_{ref} \leftarrow \text{Upsample}(\text{VAE}_{decode}(z_0^{lr}))$ ; // Decode and upsample in pixel domain
$z_0^{ref} \leftarrow \text{VAE}_{encode}(I_{ref})$ ;                      // Final clean reference latent
```
// Stage 2:  Guided High-Resolution Denoising
```
$c_{lq} \leftarrow \text{VAE}_{encode}(I_{lq})$ ;        // Encode UHD image as the main condition
$z_T \sim \mathcal{N}(0, I)$ ;          // Initialize high-resolution Gaussian noise
**for** $t = T$ **to** 1 **do**
$\quad$ $z_t^{ref} \leftarrow \text{add\_noise}(z_0^{ref}, z_T, t)$ ; // Add corresponding noise to reference
$\quad$ $P_t \leftarrow \text{CropPatches}(z_t)$ ;              // Crop current latent into patches
$\quad$ $C_{lq} \leftarrow \text{CropPatches}(c_{lq})$ ;        // Crop condition into corresponding
$\quad$ patches
$\quad$ $P_{t-1} \leftarrow \emptyset$ ;              // Initialize set for denoised patches
$\quad$ **foreach** patch $(p_t, c_p) \in (P_t, C_{lq})$ **do**
$\quad\quad$ $p_t^{ref} \leftarrow \text{GetCorrespondingPatch}(z_t^{ref})$ ;          // Get reference patch
$\quad\quad$ $p_t' \leftarrow G_{\text{freq}}(p_t, p_t^{ref})$ ;                      // Apply Frequency Guidance
$\quad\quad$ $p_{t-1} \leftarrow \epsilon_\theta(p_t', t, c_p)$ ;    // Denoise patch with condition and $G_{\text{feat}}$
$\quad\quad$ inside U-Net
$\quad\quad\lfloor\ P_{t-1} \leftarrow P_{t-1} \cup \{p_{t-1}\}$
$\quad\lfloor\ z_{t-1} \leftarrow \text{StitchPatches}(P_{t-1})$ ;        // Stitch patches back with blending
```
// Stage 3:  Reconstruction
```
$I_{rec} \leftarrow \text{VAE}_{decode}/\text{VAE-FT}(z_0)$ ;      // Decode with standard or fine-tuned
VAE
**return** $I_{rec}$

---

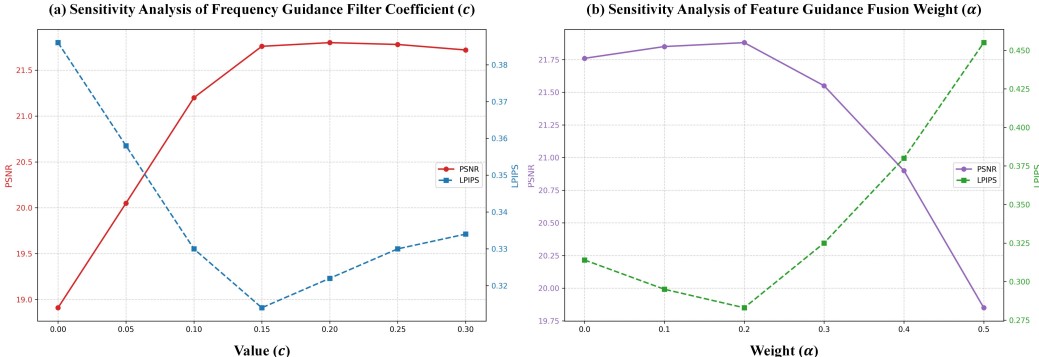

Figure 8: Hyperparameter sensitivity analysis on the UHD-LL dataset.

# B  HYPERPARAMETER ANALYSIS

To evaluate the sensitivity of the introduced hyperparameters, we conducted a systematic analysis on the UHD-LL dataset. Our study focuses on the two key parameters governing the proposed synergistic guidance mechanism: the filter coefficient $c$ in FreqG and the fusion weight $\alpha$ in FeatG. Figure 8 provides an overview of the evaluation results, from which it can be observed that FreeAdapt maintains strong robustness across a broad parameter range. The default configuration ($c = 0.15$, $\alpha = 0.20$) achieves consistently favorable performance across different tasks and backbone models, indicating that these settings fall within a stable operating regime and do not require extensive tuning.

**Analysis of Frequency Guidance Filter Coefficient** ($c$). For the Frequency Guidance module, the coefficient $c$ determines the bandwidth of the low-pass filter responsible for stabilizing global phase information. In Figure 8(a), the curves illustrate that when $c < 0.10$, the constraint on low-frequency structure becomes insufficient, occasionally leading to color shifts or structural inconsistencies across patches. Conversely, values above $0.20$ restrict the flexibility of the diffusion process, resulting in a slight reduction in texture diversity. Within the interval $c \in [0.10, 0.20]$, however, the performance curves remain nearly flat, demonstrating that the frequency-domain guidance is intrinsically robust to moderate changes in the filter bandwidth.

**Analysis of Feature Guidance Fusion Weight** ($\alpha$). For the Feature Guidance module, the fusion weight $\alpha$ controls the strength of global context injected into local patches. The trend in Figure 8(b) shows a clear trade-off between artifact suppression and detail preservation as $\alpha$ varies from 0.0 to 0.5. Under-guidance ($\alpha < 0.10$) fails to adequately suppress high-frequency hallucinations in smooth regions, which negatively affects perceptual quality despite reasonable PSNR values. Over-guidance ($\alpha > 0.30$) places excessive emphasis on low-resolution reference features, leading to over-smoothing and a noticeable degradation in both PSNR and LPIPS. A stable performance plateau emerges within $\alpha \in [0.15, 0.25]$, where $\alpha = 0.20$ offers the most balanced trade-off between structural consistency and texture fidelity.

**Correlation with Upscaling Factor.** We further examine the relationship between hyperparameter choices and the upscaling factor. When restoring images upsampled to 8K resolution, patch-based inference faces increased uncertainty due to larger spatial gaps, and slightly higher values of $c$ and $\alpha$ help anchor global structure and suppress hallucinations. For smaller upscaling ratios, patches inherently preserve more structural information, and correspondingly smaller settings of these parameters allow the model to focus more effectively on refining high-frequency details. Despite this correlation, the default configuration ($c = 0.15, \alpha = 0.20$) proves consistently robust across standard UHD restoration scenarios, eliminating the need for task-specific tuning.

# C  LIMITATIONS AND FUTURE WORK

Despite the encouraging results achieved by FreeAdapt, our method still has certain limitations, which also point to promising directions for future research.

**Computational Efficiency**. The primary limitation of our approach lies in its computational and time cost. As an iterative denoising framework, restoring a UHD image requires performing dozens of denoising steps, which may become a bottleneck in scenarios demanding fast responses. Although our method is training-free and therefore saves substantial resources during adaptation, the inference speed still has room for improvement. A highly promising direction is to distill our synergistic guidance mechanism into one-step or few-step generative models. Through knowledge distillation, we expect to retain high restoration quality while improving inference efficiency by an order of magnitude, making the method more practical for real-world applications.

**Practical Applicability and Use Cases**. While LDMs+FreeAdapt is not optimized for real-time or edge-side deployment, it is highly suitable for offline UHD restoration workflows where perceptual fidelity takes precedence over computational cost. In professional film and television remastering, for instance, frame-level processing is typically performed offline, and longer runtimes are acceptable for achieving visually coherent and artifact-free results. In the digital preservation of cultural heritage, the ability to recover nuanced textures and subtle structural details is far more critical than processing speed, making LDMs+FreeAdapt a favorable choice. Similarly, large-scale remote sensing pipelines routinely handle extremely high-resolution satellite imagery in batch mode, where offline processing and high-quality reconstruction directly benefit downstream tasks. These examples highlight that, despite its iterative nature, LDMs+FreeAdapt provides clear practical value in domains where quality-centric restoration is essential, and future acceleration efforts may further broaden its applicability.

**Benchmark Limitations and Generalization to Real-World Data.** Current UHD-IR benchmarks rely on synthetic degradation pipelines because collecting paired real-world UHD data remains extremely challenging. We acknowledge that this limits the diversity and realism of degradation patterns. As high-quality UHD datasets continue to emerge, we consider extending evaluation to real-world benchmarks an important direction for future work. This will allow a more comprehensive assessment of FreeAdapt's generalization capability beyond synthetic settings and will help validate its robustness under practical, unconstrained degradations.

**Architectural Adaptability**. The current FFSG design primarily targets U-Net based latent diffusion models and their extensions such as ControlNet (Zhang et al., 2023). In this setting, FeatG is integrated into the self-attention layers of the decoder, and FreqG benefits from the hierarchical multi-scale feature processing of the U-Net. Recent progress in diffusion modeling has, however, shifted toward Transformer-based architectures, most notably Diffusion Transformers (DiTs), which now form the backbone of many state-of-the-art diffusion systems. Although the present implementation is built upon U-Net structures, the underlying principles of FFSG are not inherently tied to this architecture, and both guidance components can be adapted to DiTs with appropriate modifications.

FreqG is fundamentally architecture-agnostic because it operates directly on the latent representation before the denoising step. The fused latent can be passed through the standard patchification and tokenization stages of a DiT model without changes to the backbone, which makes the frequency-domain guidance naturally compatible with Transformer-based diffusion architectures. FeatG, in contrast, requires adaptation since its current form relies on U-Net style self-attention. A feasible extension is to introduce a modified DiT block in which the local self-attention computed on high-resolution patch tokens is complemented by a global cross-attention mechanism that attends to reference tokens extracted from the low-resolution guidance image. The outputs of these two attention paths are subsequently combined using the same linear blending strategy described in Eq. 9, after which the resulting tokens are processed by the standard feed-forward layers of the DiT block. This formulation preserves the core objective of FeatG, namely maintaining local detail fidelity while injecting global contextual cues, yet aligns it with the token-based computation of DiT models.

## D  COMPUTATIONAL EFFICIENCY ANALYSIS

We provide a quantitative analysis of the computational efficiency of FreeAdapt in Table 5, which reports parameters, FLOPs, peak GPU memory usage, and inference latency for restoring a single 4K image ($3840 \times 2160$). Although diffusion-based UHD restoration is inherently more computationally demanding than regression-based architectures, FreeAdapt introduces only a marginal overhead compared with the baseline patch-based inference pipeline. For example, LDM-Ours increases

Table 5: Quantitative comparison of computational efficiency for restoring a single 4K image. PI denotes patch-based inference, and DF denotes DemoFusion.

| Model | Params | FLOPs | VRAM | Latency |
|---|---|---|---|---|
| ERR | 1.31 M | 307.52 G | 7.11 GB | 0.63 s |
| UHDformer | 0.34 M | 399.01 G | 12.65 GB | 0.60 s |
| Wave-Mamba | 1.26 M | 948.76 G | 14.54 GB | 1.36 s |
| UHDFour | 32.08 M | 596.13 G | 2.48 GB | 0.22 s |
| SwinIR | 15.06 M | 2764.31 G | 6.85 GB | 1.01 s |
| DreamUHD | 1.16 M | 496.44 G | 14.65 GB | 0.56 s |
| FFTformer-PI | 15.79 M | 58.93 T | 3.60 GB | 51.79 s |
| Restormer-PI | 19.94 M | 46.60 T | 2.51 GB | 28.69 s |
| Uformer-PI | 50.88 M | 38.43 T | 2.31 GB | 13.79 s |
| LDM-PI | 1243.19 M | 954.62 T | 6.23 GB | 178.69 s |
| StableSR-PI | 1295.69 M | 1112.82 T | 7.07 GB | 212.32 s |
| DiffBIR-PI | 1596.53 M | 1233.20 T | 7.27 GB | 228.74 s |
| LDM-DF | 1243.19 M | 1236.31 T | 6.26 GB | 231.57 s |
| StableSR-DF | 1295.69 M | 1442.54 T | 7.10 GB | 278.09 s |
| DiffBIR-DF | 1596.53 M | 1718.30 T | 7.31 GB | 298.70 s |
| LDM-Ours | 1243.19 M | 1023.10 T | 6.33 GB | 192.67 s |
| StableSR-Ours | 1295.69 M | 1192.62 T | 7.21 GB | 232.93 s |
| DiffBIR-Ours | 1596.53 M | 1408.88 T | 7.39 GB | 247.03 s |

FLOPs by approximately 7% relative to LDM-PI, yet yields substantially improved perceptual quality. This cost–performance ratio highlights the efficiency of our guidance mechanism.

The overall computational load arises from three inherent properties of diffusion-based UHD restoration: the iterative nature of diffusion sampling, the substantial parameter count of pre-trained generative backbones, and the redundancy introduced by patch-based inference, where overlapping patches must be processed repeatedly to avoid boundary artifacts. Despite these sources of overhead, FreeAdapt avoids additional multi-scale stages or multi-pass fusion, enabling it to remain more efficient than methods such as DemoFusion while consistently achieving higher restoration fidelity.

Future work will investigate techniques for further reducing inference complexity, including step distillation, architectural acceleration, and low-bit quantization, to make high-fidelity UHD restoration increasingly practical in resource-limited environments.

## E UHD IMAGE DERAINING RESULTS

To comprehensively evaluate the applicability of our method to UHD-IR tasks, we additionally conduct experiments on a UHD image deraining dataset, 4K-Rain13k (Chen et al., 2024). We compare FreeAdapt-equipped diffusion models (LDM, StableSR, and DiffBIR) against state-of-the-art deraining and UHD restoration approaches, including Uformer (Wang et al., 2022), Restormer (Zamir et al., 2022), SwinIR (Liang et al., 2021), UDR-Mixer (Chen et al., 2024), UDR-S2Former (Chen et al., 2023), DreamUHD (Liu et al., 2025b), and ERR (Zhao et al., 2025).

Table 6 summarizes the quantitative results on the 4K-Rain13k dataset. Traditional deraining models (e.g., Uformer and Restormer) achieve strong performance on distortion-oriented metrics, but diffusion-based variants equipped with FreeAdapt show competitive or superior perceptual quality. In particular, DiffBIR-Ours achieves the best or second-best scores on LPIPS, DISTS, CLIP-IQA, MUSIQ, and MANIQA, demonstrating its strong capability in preserving realistic textures and producing perceptually faithful outputs under heavy rain degradation. In addition, Figure 9 provides qualitative comparisons on UHD rainy images. Existing approaches often produce over-smoothed regions, residual streak artifacts, or inconsistent textures when encountering complex rain patterns. In contrast, diffusion backbones equipped with FreeAdapt generate clearer structures, better-preserved textures, and more coherent details, yielding visually stable restoration results

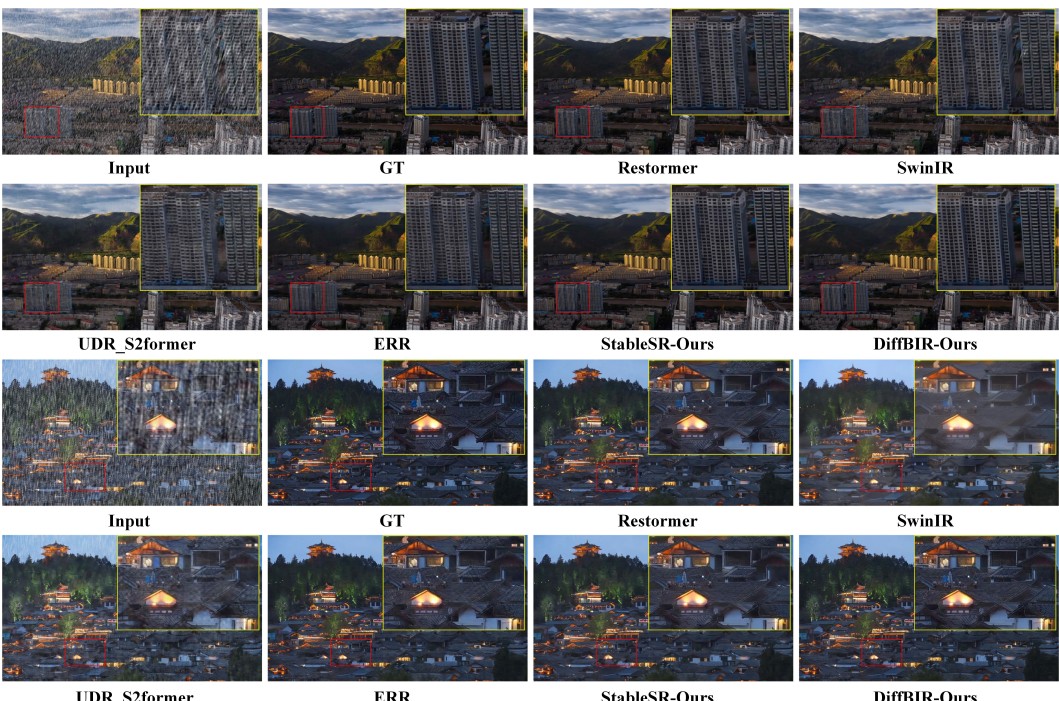

Figure 9: Visual comparison between FreeAdapt and state-of-the-art UHD-IR methods on the 4K-Rain13k dataset.

Table 6: Quantitative comparison on 4K-Rain13k dataset (Chen et al., 2024). The symbols ↑ and ↓ respectively represent that higher or lower values indicate better performance. Bold represents the best and underline represents the second best.

| Method | PSNR ↑ | SSIM ↑ | LPIPS ↓ | DISTS ↓ | CLIPIQA ↑ | MUSIQ ↑ | MANIQA ↑ |
|---|---|---|---|---|---|---|---|
| Uformer | **34.67** | 0.953 | 0.120 | 0.072 | 0.319 | 33.44 | 0.256 |
| Restormer | 34.09 | **0.955** | 0.132 | 0.088 | 0.304 | 33.51 | 0.256 |
| Swinir | 31.57 | 0.940 | 0.126 | 0.070 | 0.229 | 32.37 | 0.236 |
| UDR-Mixer | 34.28 | 0.950 | 0.133 | 0.086 | 0.344 | 33.17 | 0.255 |
| UDR-S2former | 27.43 | 0.913 | 0.199 | 0.105 | 0.358 | 29.64 | 0.228 |
| DreamUHD | 29.26 | 0.912 | 0.221 | 0.131 | 0.241 | 30.07 | 0.206 |
| ERR | 34.43 | 0.951 | 0.120 | 0.070 | 0.366 | 33.60 | 0.261 |
| LDM-Ours | 25.89 | 0.900 | 0.174 | 0.066 | 0.379 | 34.73 | 0.283 |
| StableSR-Ours | 28.04 | 0.912 | 0.157 | 0.056 | 0.379 | 34.23 | 0.277 |
| DiffBIR-Ours | 29.11 | 0.937 | **0.111** | **0.039** | **0.384** | **34.98** | **0.284** |

across diverse scenes. These findings confirm that FreeAdapt generalizes well to the UHD deraining task and maintains high perceptual fidelity across both quantitative and qualitative dimensions.

## F  MORE EXPERIMENTAL RESULTS

To further demonstrate the superiority of our approach, we provide additional full-resolution qualitative comparisons in Figure 10, Figure 11 and Figure 12 on the UHD-LL (Li et al., 2023a), UHD-Haze (Wang et al., 2024a), and UHD-Blur (Wang et al., 2024a) tasks, respectively. These results clearly show that, compared with other state-of-the-art methods, our method consistently achieves advantages in recovering fine textures, suppressing artifacts, and preserving color fidelity.

Figure 13, Figure 14 and Figure 15 present visual comparisons between our FreeAdapt framework and other diffusion adaptation methods on two different backbones, LDM and DiffBIR, across the

UHD-LL, UHD-Haze, and UHD-Blur tasks. The results show that methods such as MultiDiffusion and DemoFusion often produce blurry outputs, repeated contents, or unnatural textures when handling complex scenes. In contrast, our method, through effective synergistic guidance, consistently generates structurally correct and detail-coherent high-quality images.

## G  STATEMENT ON THE USE OF LARGE LANGUAGE MODELS

In the preparation of this paper, we used large language models (LLMs) as an auxiliary tool. The primary role of LLMs was to assist with language polishing, phrasing improvements, grammar checking, and enhancing the clarity and fluency of certain sentences, ensuring that our research ideas and technical details are expressed more accurately and professionally. Importantly, LLMs were not involved in the core aspects of this work, including research conception, experimental design, result generation, or data analysis.

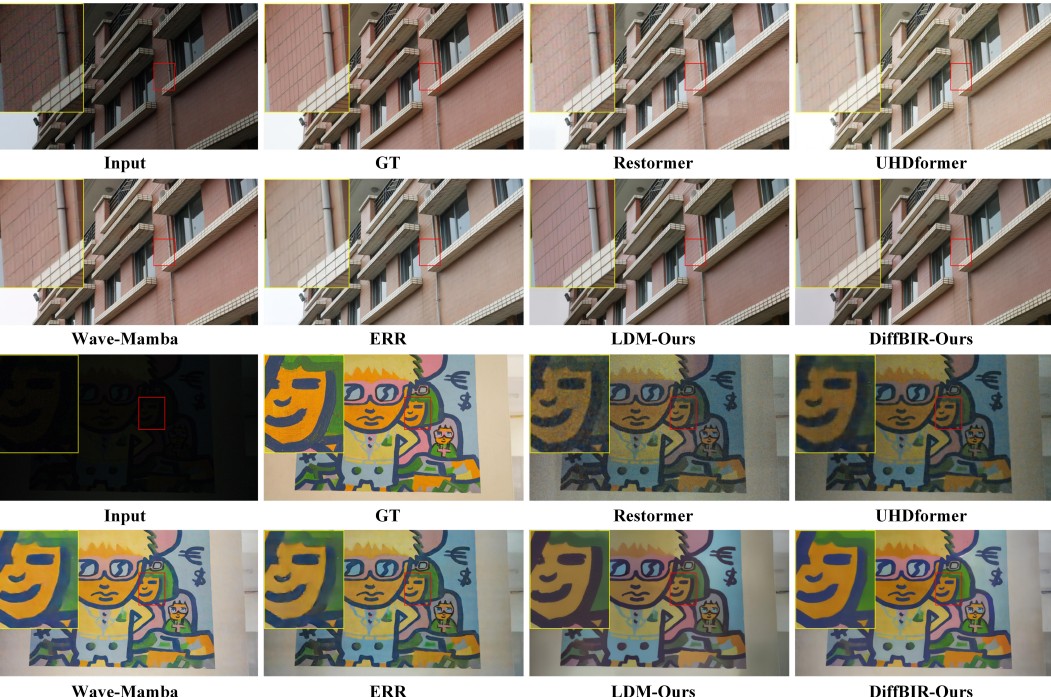

Figure 10: Visual comparison between FreeAdapt and state-of-the-art UHD-IR methods on the UHD-LL dataset.

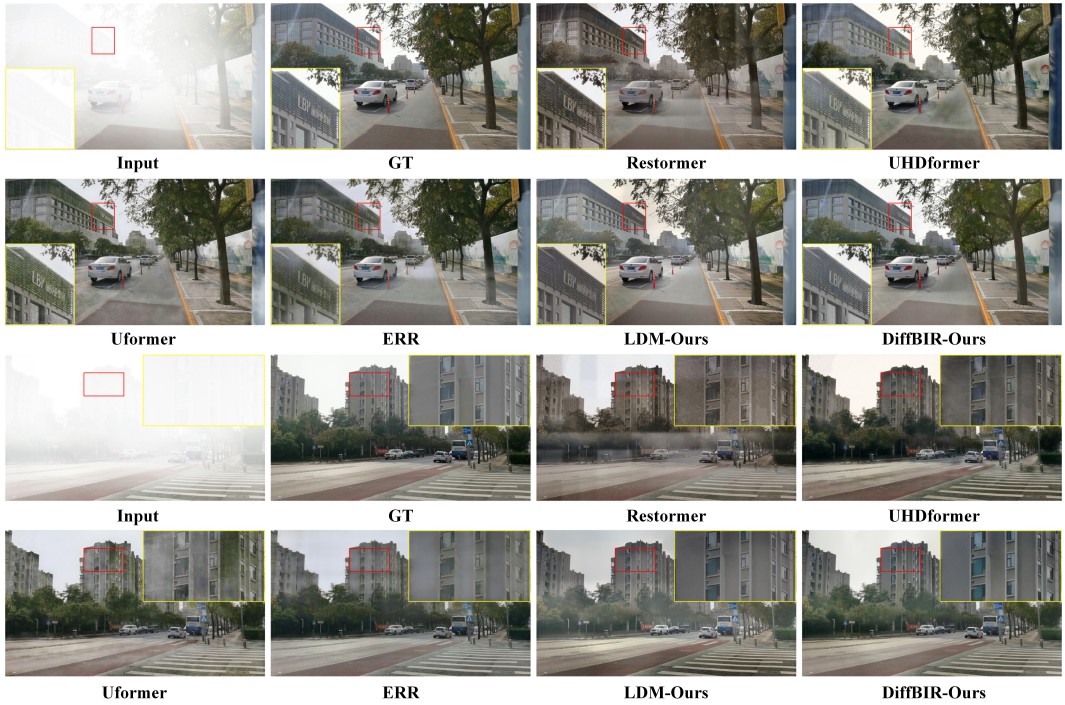

Figure 11: Visual comparison between FreeAdapt and state-of-the-art UHD-IR methods on the UHD-Haze dataset.

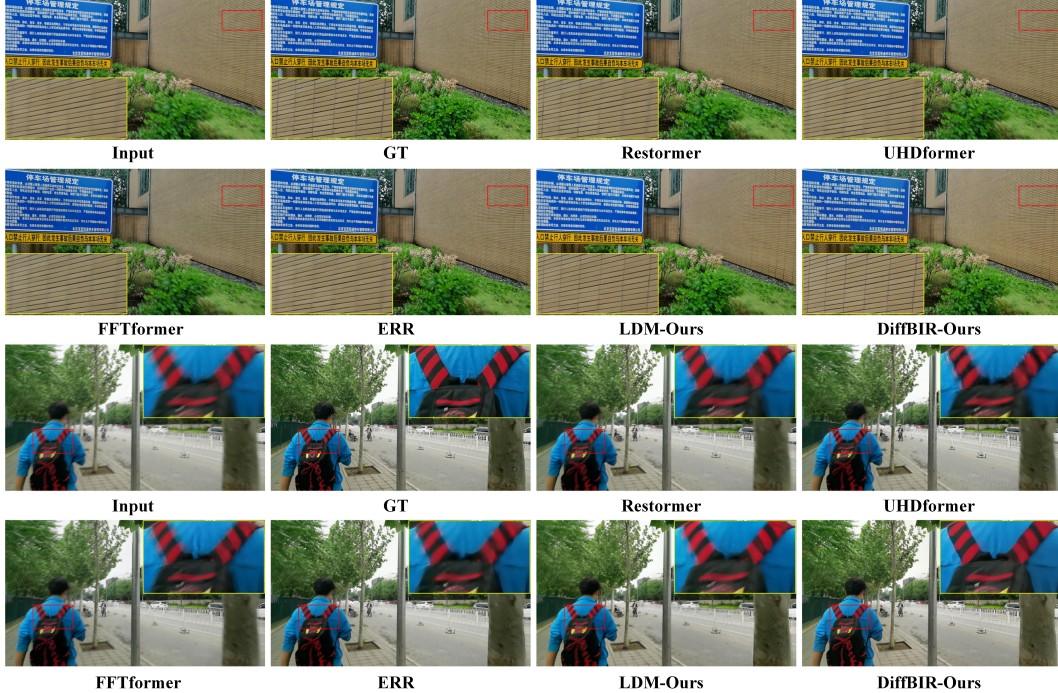

Figure 12: Visual comparison between FreeAdapt and state-of-the-art UHD-IR methods on the UHD-Blur dataset.

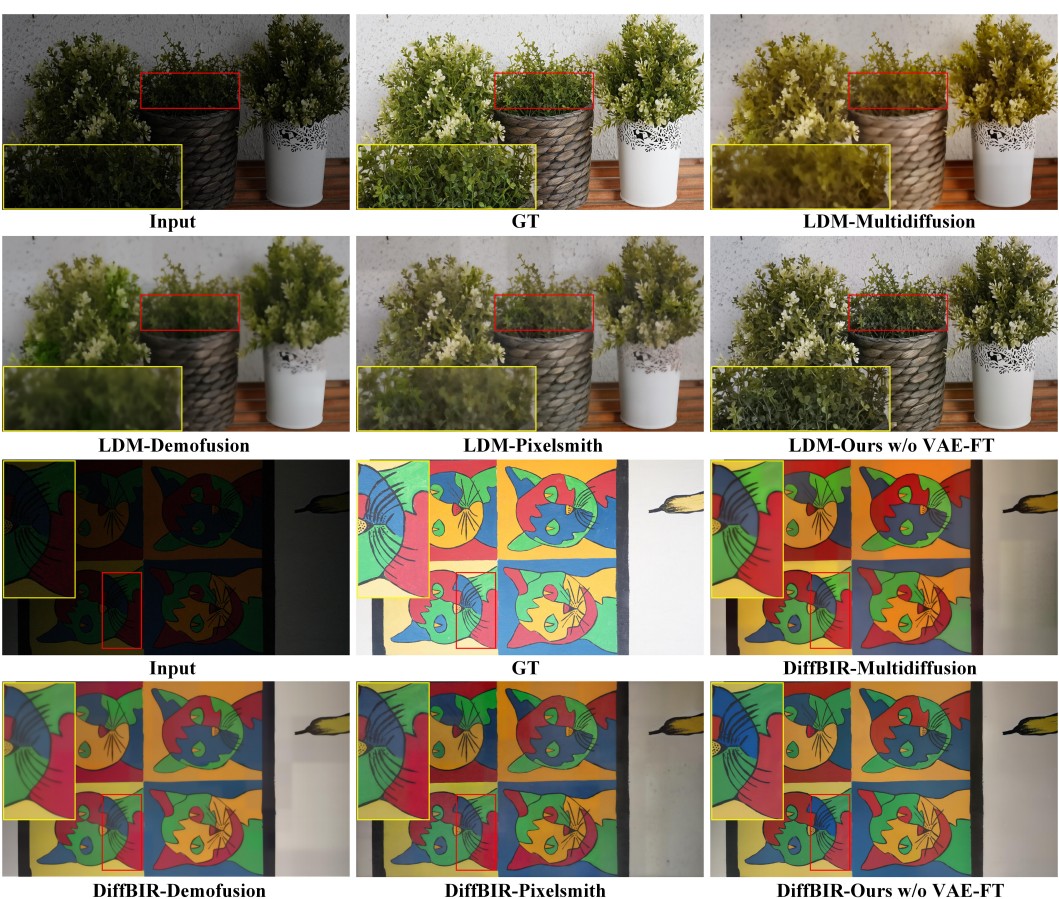

Figure 13: Visual comparison of FreeAdapt against training-free diffusion-based adaptation methods on the UHD-LL dataset.

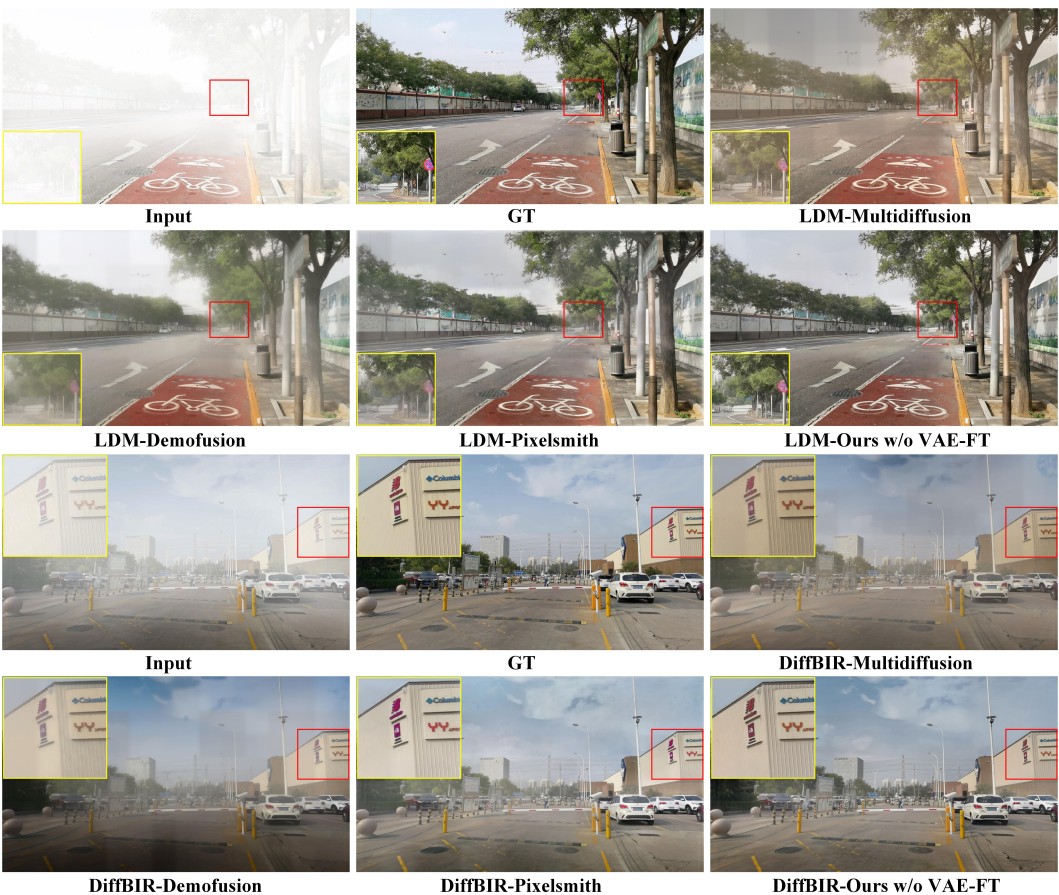

Figure 14: Visual comparison of FreeAdapt against training-free diffusion-based adaptation methods on the UHD-Haze dataset.

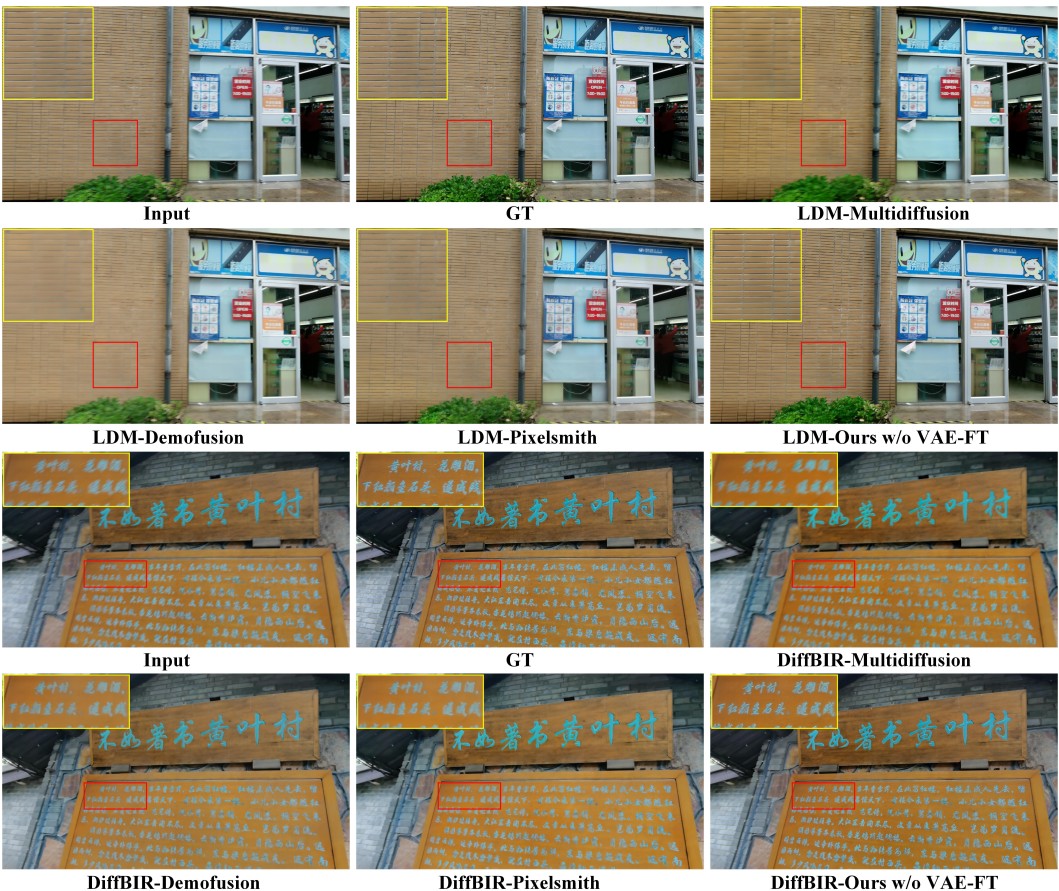

Figure 15: Visual comparison of FreeAdapt against training-free diffusion-based adaptation methods on the UHD-Blur dataset.

