# OpenReview forum: "FreeAdapt: Unleashing Diffusion Priors for Ultra-High-Definition Image Restoration"
_ICLR.cc/2026/Conference — ICLR 2026 Poster_

### Official Review · Reviewer_PoKA · 2025-10-31

**Soundness:** 2
**Presentation:** 3
**Contribution:** 2
**Rating:** 4
**Confidence:** 5

**Summary:**

The paper proposes FreeAdapt, a plug-and-play framework designed to enhance ultra-high-definition image restoration (UHD-IR) using diffusion priors. To address the global inconsistency and detail loss issues of existing latent diffusion models (LDMs) caused by patch-based inference and VAE bottlenecks, the paper introduce a training-free Frequency Feature Synergistic Guidance  mechanism that provides guidance at each denoising step. FFSG includes two modules: Frequency Guidance, which fuses phase information from a reference image in the frequency domain to ensure structural consistency, and Feature Guidance (FeatG), which injects global contextual information into U-Net self-attention layers to suppress artifacts and preserve fine details. Additionally, an optional VAE fine-tuning module with skip connections is proposed to further improve texture reconstruction. Experiments show that FreeAdapt achieves superior quantitative and qualitative performance compared to existing UHD-IR and diffusion-based adaptation methods.

**Strengths:**

1. It achieves promising results in terms of perceptual quality metrics.

2. Exploring the application of diffusion models in UHD image restoration is an interesting direction.

**Weaknesses:**

1. The second claimed contribution — the frequency and feature guidance modules — appears highly similar to the approach proposed in [1]. Please clearly describe the differences between your method and [1]. If there are no substantial distinctions, these components should not be presented as core contributions.

2. How does the proposed method compare in terms of computational efficiency with existing UHD-IR approaches? Although I understand this may be a limitation of the current method, the paper does not provide any relevant data. Please report a quantitative comparison including GPU memory usage, inference speed, number of parameters, and FLOPs.

3. Applying diffusion models to UHD image restoration incurs substantial computational costs, making deployment on edge devices challenging. Compared with existing methods such as UHDformer and ERR, the practical applicability and significance of the proposed approach remain unclear. A discussion of its real-world deployment potential, efficiency trade-offs, and specific use cases is necessary.

4. The paper lacks comparative results on UHD image deraining.

[1] FAM Diffusion: Frequency and Attention Modulation for High-Resolution Image Generation with Stable Diffusion. CVPR 2025.

**Questions:**

If the authors can address the aforementioned issues, I would consider raising my evaluation score.

---

> ### Author Response · Authors · 2025-11-21
> **Response to Reviewer PoKA (part 1/2)**
>
> Thank you for your insightful review and valuable feedback. We address your questions in turn below:
>
> >`w1: Distinctions Between FreeAdapt and FAM Diffusion`
>
>
> We sincerely thank the reviewer for raising this important question. Although both FAM Diffusion [1] and our method involve frequency-domain and attention-related guidance, their motivations and mechanisms differ fundamentally.
>
> **Difference of Frequency Fusion Mechanism.** FM replaces low-frequency coefficients, whereas FreqG injects phase information. In FAM Diffusion, FM operates directly on the Fourier-transformed latent by retaining the high-frequency coefficients of the high-resolution latent $z_t$ and replacing its low-frequency coefficients with those from a diffused low-resolution latent $\tilde{z}_t$. However, when applied to UHD restoration, this coefficient-level substitution tends to introduce global color shifts, as illustrated in Figure 6 of the paper. In contrast, FreqG decomposes the Fourier representation into amplitude and phase, preserves the amplitude $A_t$ of the current latent, and selectively injects the reference phase $\phi_t^{\text{ref}}$, which more reliably maintains global structural consistency.
>
> **Difference of Attention Modulation Strategy.**  AM blends attention matrices, whereas FeatG blends attention outputs. In FAM Diffusion, AM upsamples the low-resolution attention matrix $M^{n}$ and fuses it with the high-resolution matrix $M^{m}$ before value aggregation, a process that is difficult to align with patch-based UHD restoration. In contrast, FeatG adopts a dual-branch strategy that blends the attention results rather than the matrices themselves. Specifically, FeatG computes the standard self-attention output $Attn_{local}$ from the current patch features, and in parallel constructs a global branch by sampling the reference query $Q^{ref}$, cropping it to the corresponding region, and upsampling it before attending to the reference keys and values $(K^{ref}, V^{ref})$ to obtain $Attn_{global}$. The two attention outputs are then linearly combined, allowing global semantics to guide the restoration while preserving the detailed structures captured by the local branch.
>
> **Difference of resolution handling.** FAM Diffusion processes the entire high-resolution latent in one pass, whereas FreeAdapt operates in a patch-based manner.  FAM Diffusion performs diffusion and attention on a full-resolution latent, which requires constructing complete attention matrices whose memory usage grows quadratically with image size. This makes AM extremely memory-intensive and impractical on consumer GPUs for 4K/8K resolutions. In contrast, FreeAdapt is designed around patch-based inference, enabling FFSG to run on ultra-high-resolution inputs with manageable memory consumption and without computing full-resolution attention.
>
> **Difference of task objective.** FAM Diffusion targets image generation, whereas FreeAdapt targets image restoration.  FAM Diffusion focuses on high-resolution image generation, where the model synthesizes new content from prompts and aims for coherent global layout and consistent local textures. FreeAdapt, however, addresses UHD image restoration, which requires strict input–output fidelity: the restored result must preserve the geometry and fine details of the degraded input.
>
> In summary, the motivations, mechanisms, and constraints of FM/AM in FAM Diffusion differ fundamentally from FreqG/FeatG in FreeAdapt, which are specifically developed for UHD image restoration rather than generation.
>
> [1] FAM Diffusion: Frequency and Attention Modulation for High-Resolution Image Generation with Stable Diffusion. CVPR 2025.
>
> >`w2: Computational Efficiency Comparison`
>
> We thank the reviewer for this valuable suggestion. To provide a clear and quantitative assessment, we report the number of parameters, FLOPs, GPU memory usage, and inference time for restoring a single 4K image in **Appendix D**. FreeAdapt introduces only a marginal computational overhead relative to the standard patch-based inference pipeline. For example, LDM-Ours requires approximately 7% more FLOPs than LDM-PI, yet delivers a substantial improvement in perceptual quality. This favorable cost–performance trade-off reflects the lightweight nature of our guidance mechanism.
>
> The overall computational burden of diffusion-based UHD restoration primarily arises from three inherent factors: (1) the iterative sampling process required by diffusion models, (2) the large parameter count of pre-trained generative backbones, and (3) the redundancy introduced by patch-based inference, in which overlapping patches must be evaluated multiple times to ensure smooth boundaries at 4K resolution. Future work will investigate strategies for further improving computational efficiency, including step distillation, architectural acceleration, and low-bit quantization, to make high-fidelity diffusion-based UHD restoration more practical in resource-limited environments.

---

> ### Author Response · Authors · 2025-11-21
> **Response to Reviewer PoKA (part 2/2)**
>
> >`w3: Real-world applicability and efficiency trade-offs`
>
> We thank the reviewer for raising this important and practical concern. We fully agree that diffusion-based UHD restoration incurs noticeable computational overhead, as acknowledged in **Section 5 and Appendix C**. LDMs+FreeAdapt is not intended for edge devices or real-time scenarios. Our method is fundamentally a quality-first rather than speed-first solution.
>
> Regarding efficiency–quality trade-offs, existing feed-forward UHD restoration models (e.g., UHDformer, ERR) are highly efficient but often produce overly smoothed results and struggle with complex real-world degradations. In contrast, LDMs+FreeAdapt achieves significantly higher perceptual fidelity, especially in UHD settings where structural realism and fine texture recovery are essential. Although iterative denoising is more computationally demanding, it enables a level of restoration quality that feed-forward networks currently cannot match, making our approach suitable for use cases where quality outweighs runtime constraints.
>
> Regarding real-world applicability, several practical domains naturally accommodate offline, quality-centric UHD restoration:
> (1) Film and TV remastering: high-quality frame-by-frame restoration of old films or archival footage routinely tolerates long processing times, as visual fidelity is prioritized over speed in professional pipelines.
> (2) Digital cultural heritage restoration: artwork, mural, and historical photo scans require accurate texture recovery and artifact removal; these tasks are typically processed offline on dedicated servers, making quality the dominant factor.
> (3) Large-scale remote sensing imagery: satellite images generally have very high spatial resolution and are almost always processed offline; thus, computationally intensive but high-quality restoration is acceptable and often preferred.
>
> Finally, we have expanded the discussion in Sec. 5 to explicitly include these use cases and clarify the deployment scenarios in which FreeAdapt is most relevant. Improving computational efficiency remains an important direction for our future work. We are actively exploring distillation-based one- or few-step variants to significantly reduce runtime while retaining the strong restoration performance enabled by our guidance mechanism.
> We sincerely appreciate the reviewer’s suggestion and have incorporated the requested discussion into the revised manuscript.
>
> >`w4: Comparisons for UHD image deraining`
>
> We thank the reviewer for highlighting the lack of UHD deraining experiments. In the revised manuscript, we have added a dedicated evaluation on the 4K-Rain13k dataset in the **Appendix E**. The new results include comparisons between FreeAdapt-equipped diffusion backbones (LDM, StableSR, and DiffBIR) and state-of-the-art deraining and UHD-IR methods such as Uformer, Restormer, SwinIR, UDR-Mixer, UDR-S2Former, DreamUHD, and ERR.
>
> **Table 6:** Quantitative Comparison on 4K-Rain13k Dataset. (**Excerpted**)
>
> | **Method**    | **PSNR ↑** | **LPIPS ↓** | **DISTS ↓** | **MUSIQ ↑** |
> | ------------- | ---------- | ----------- | ----------- | ----------- |
> | Uformer       | **34.67**  | *0.120*     | 0.072       | 33.44       |
> | Restormer     | 34.09      | 0.132       | 0.088       | 33.51       |
> | UDR-Mixer     | 34.28      | 0.133       | 0.086       | 33.17       |
> | UDR-S2former  | 27.43      | 0.199       | 0.105       | 29.64       |
> | DreamUHD      | 29.26      | 0.221       | 0.131       | 30.07       |
> | ERR           | *34.43*    | *0.120*     | 0.070       | 33.60       |
> | StableSR-Ours | 28.04      | 0.157       | *0.056*     | *34.23*     |
> | DiffBIR-Ours  | 29.11      | **0.111**   | **0.039**   | **34.98**   |
>
> The quantitative results in **Table 6** show that diffusion backbones enhanced with FreeAdapt achieve competitive or superior perceptual performance, with DiffBIR-Ours obtaining the best or second-best scores across most perceptual metrics. Complementary qualitative comparisons in **Figure 9** demonstrate that our method produces clearer structures and more coherent textures than existing approaches, which often suffer from residual streaks or over-smoothing under heavy rain. These newly added experiments confirm that FreeAdapt generalizes well to UHD deraining and maintains strong perceptual fidelity in both quantitative and qualitative evaluations.

---

> > ### Comment · Reviewer_PoKA · 2025-11-28
> >
> > Thank you very much for the detailed response, which has addressed most of my concerns. I have decided to raise my score to 6. Since OpenReview currently does not allow score updates, I will update it once the system permits, or I will inform the AC directly.

---

> ### Author Response · Authors · 2025-11-28
> **Official Response by Authors**
>
> Thank you very much for your thoughtful follow-up and for taking the time to reconsider your assessment. We sincerely appreciate your constructive feedback throughout the review process, which has greatly helped us improve the clarity and quality of the manuscript.
>
> Thank you again for your effort and for updating the score. We truly appreciate your support, and we are happy to discuss any further questions or suggestions.

---

### Official Review · Reviewer_H5xg · 2025-10-31

**Soundness:** 3
**Presentation:** 3
**Contribution:** 2
**Rating:** 4
**Confidence:** 3

**Summary:**

This paper introduces a training-free method FreeAdapt to utilize pre-trained diffusion priors for ultra-high-definition image restoration. The core of the framework is a training-free frequency-feature synergistic guidance mechanism, which enforces global structural consistency through frequency guidance and suppresses artifacts while preserving local detail fidelity via feature guidance. By fine-tuning VAE decoder, it further enhances the reconstruction of texture and detail. Extensive experiments demonstrate the effectiveness of the proposed method.

**Strengths:**

● The proposed frequency and feature guidance strategies effectively resolves the structure inconsistencies and artifacts in patch-based inference results.
● By fine-tuning VAE decoder, it further enhances the reconstructed texture and mitigates the lossy compression characteristic.
● Extensive experiments demonstrate the effectiveness of the proposed method on three synthetic benchmark.

**Weaknesses:**

● Training the VAE decoder conflicts with the authors' claim of ``training-free approach''. In fact, in Fig. 3, using only FreqG/FeatG yields only marginal performance gains. I highly doubt the validity of the restoration results, which stems from fine-tuning the VAE decoder network across three synthetic degradation scenarios.
● The part of ``reference image generation'' is confusing. Transforming a degraded image to a clean image require performing sdedit process or utizling a pre-trained I2I backbone. The authors need to provide more details about it.
● The testing scenarios are limited. Current benchmark are constructed by applying several synthetic degradation factors, which can be fitted by only fine-tuning the decoder network with the corresponding datasets. The authors are suggested to perform experiments on AI-generated or real-world benchmark.

**Questions:**

● More experiments need to be conducted to demonstrate the proposed ``training-free'' approaches without fine-tuning decoder network.
● The authors are recommended to provide image results in supplementary material.

**Details Of Ethics Concerns:**

None.

---

> ### Author Response · Authors · 2025-11-21
> **Response to Reviewer H5xg (part 1/2)**
>
> Thank you for your comprehensive and insightful review. We address your questions in turn below:
>
> >`w1：Questioning training-free claim and attributing gains to VAE fine-tuning`
>
> We sincerely apologize for the confusion caused by our description of “training-free” and “VAE fine-tuning.” We clarify that the substantial performance improvements of our method originate from the core, fully training-free FFSG module, while the VAE-FT component is an optional, task-level auxiliary used only to mitigate the known limitations of standard VAEs in latent diffusion models. Below we address the reviewer’s concerns in detail.
>
> **Role of the VAE in latent diffusion models.** In latent diffusion frameworks, the VAE is responsible for encoding images into a latent space. However, the standard VAE introduces an information bottleneck that inevitably removes high-frequency details, which becomes particularly problematic for UHD restoration tasks.
>
> **How VAE FT is trained and what it learns.**  The goal of VAE FT is not to adapt to any specific synthetic degradation but to compensate for the detail loss introduced by the encoding process. During training, a low quality image and its corresponding high quality image are both passed through the VAE encoder. This produces the high quality latent as well as residual features extracted from the low quality input. The decoder receives the high quality latent together with the residual features through skip connections, which provide structural cues that assist in reconstructing details lost during encoding. In this way, VAE FT learns a task level detail reconstruction prior, such as how to recover texture under low light or hazy conditions. It does not have the ability to perform restoration on its own and serves only as a complement to the diffusion model. Since the learned prior is task specific rather than model specific, the same VAE FT can be used across different diffusion backbones without retraining.
>
> **Performance gain of FFSG.** We appreciate the reviewer’s concern regarding the contribution of FreqG and FeatG. As shown in the newly added **Table 2** and **Table 3** in the revised manuscript, FFSG consistently accounts for the dominant performance gain . For example, on the UHD-LL task (Table 3):
>
> **Table3:** Performance Gain of FFSG on UHD-LL. (**Excerpted**)
>
> |**Method**|**PSNR ↑**|**LPIPS ↓**|
> |---|---|---|
> |**Baseline (LDM-PI)**|18.91|0.386|
> |**Baseline + FFSG (Training-Free)**|21.88|0.283|
> |_FFSG Gain_|_+2.97_|_-0.103_|
> |**Baseline + FFSG + VAE-FT (Optional)**|22.21|0.253|
> |_VAE-FT Gain_|_+0.33_|_-0.030_|
>
> These results indicate that the majority of the improvement comes from the training-free FFSG modules, while the optional VAE-FT offers an additional but comparatively smaller refinement mainly addressing detail loss introduced by the standard VAE.
>
> In summary, FFSG is fully training-free and provides the majority of the performance gain reported in our paper. VAE-FT is optional, operates only at the task level, and complements the diffusion model without altering its parameters. We hope this clarifies the role and contribution of each component.
>
>
> >`w2: Explanation of how the reference image is generated`
>
> The “reference image generation” step relies on the pre-trained I2I capability of the same diffusion backbone, and we apologize for the insufficient clarity in the original manuscript. Below we summarize the process more explicitly.
> 1. **Downsampling:** The degraded UHD image $I_{lq}$ is first downsampled to the backbone's native resolution (e.g., $512 \times 512$).
> 2. **Encoding:** This downsampled, degraded image is then passed through the VAE encoder to obtain a latent conditioning vector, $c_{lr}$.
> 3. **I2I Denoising Pass:** We feed both the initial noise $z_T^{lr}$ and the conditioning latent $c_{lr}$ into the same pre-trained backbone (LDM or DiffBIR). A single, standard reverse diffusion process is executed, yielding a clean, low-resolution latent $z_0^{lr}$. Importantly, this step does not involve SDEdit or any additional I2I network—it reuses the backbone’s inherent I2I capability.
> 4. **Upsampling and Re-encoding:** The clean latent $z_0^{lr}$ is decoded to the pixel domain, upsampled back to the target UHD resolution, and then passed through the VAE encoder again to obtain the final, resolution-aligned reference latent $z_0^{ref}$.
>
> We will revise Sec. 3.1 in the final version to make this procedure clearer and avoid any confusion.

---

> ### Author Response · Authors · 2025-11-21
> **Response to Reviewer H5xg (part 2/2)**
>
> > `w3:Evaluation on real or AI-generated benchmarks`
>
> We thank the reviewer for this constructive suggestion. The concern about the limitations of synthetic benchmarks and the possibility that VAE-FT may simply fit the synthetic degradations is very reasonable. We would like to clarify the role of VAE-FT and address the reviewer’s comments as follows.
>
> First, VAE-FT does not adapt to any specific synthetic degradation. The purpose of VAE-FT is to alleviate the inherent information–loss limitation of the base VAE. During fine-tuning, the decoder receives additional structural cues through skip connections, allowing it to recover high-frequency details that the original VAE fails to preserve. Importantly, VAE-FT itself does not perform restoration and does not learn the characteristics of any particular synthetic degradation pattern. Instead, it learns a general detail reconstruction prior that enhances the diffusion model in a task level manner. This learning process aims at improving the preservation of useful detail rather than adapting to any specific artificial corruption.
>
> Second, we agree that evaluation on real world or AI generated benchmarks would further validate the method.  Paired ultra high resolution datasets are difficult to collect, and current UHD restoration research typically relies on synthetic pipelines. Even so, we acknowledge the importance of testing under real world degradations or AI generated distortions. We are actively following the development of new UHD datasets and consider such evaluation an important direction for future work.
>
> We appreciate the reviewer’s suggestion. We will revise the manuscript to clarify the role of VAE-FT and to acknowledge the limitations of synthetic benchmarks.
>
> > `Q1: More results without VAE fine-tuning`
>
> We thank the reviewer for these helpful suggestions and are pleased to clarify that the submitted manuscript already includes the requested results. Specifically, **Table 2** presents the quantitative performance of our training free version and shows clear improvements over existing methods. **Figure 5** in the main paper, together with **Figures 13, 14 and 15** in the appendix F, provides qualitative comparisons that further demonstrate the effectiveness of FFSG without using the decoder fine tuning module. These results collectively confirm the capability of our training free approach.
>
> >`Q2: More visual results in the supplementary material`
>
> We appreciate the request for additional visual comparisons and would like to clarify that the supplementary material already provides extensive image results. Appendix F includes **Figures 10, 11 and 12**, which present full resolution and zoomed in comparisons with representative restoration baselines such as ERR and UHDformer. **Figures 13, 14 and 15** further compare our approach with other training free diffusion based methods including MultiDiffusion and DemoFusion. These visual results collectively highlight the strong structural consistency and fine detail preservation achieved by our method.

---

### Official Review · Reviewer_gDB6 · 2025-11-01

**Soundness:** 2
**Presentation:** 3
**Contribution:** 2
**Rating:** 6
**Confidence:** 4

**Summary:**

The paper addresses Ultra-High-Definition Image Restoration (UHD-IR), i.e. restoring 4K/8K images degraded by noise, blur, haze, low-light etc. The authors argue that existing UHD-IR methods (e.g. UHDFormer, ERR, Wave-Mamba) rely on architectural tricks and still hit bottlenecks due to the ill-posed nature of restoration. To overcome this, the paper proposes to leverage pretrained latent diffusion models (LDMs) as powerful generative priors. Directly applying LDMs to UHD images is problematic (patch-based inference causes stitching artifacts, loss of global context and detail due to VAE bottleneck). The core contribution is FreeAdapt, a training-free, plug-and-play framework that adapts off-the-shelf LDMs (and extensions like ControlNet) to UHD-IR. During inference, FreeAdapt applies a novel Frequency-Feature Synergistic Guidance (FFSG) mechanism and an optional VAE fine-tuning (VAE-FT). Specifically, at each denoising step, Frequency Guidance (FreqG) replaces the phase spectrum of the current latent patch with that of a low-resolution reference image, enforcing cross-patch structural consistency, while Feature Guidance (FeatG) injects global contextual features (from the reference) into the U-Net's self-attention to suppress hallucinated textures. The reference is obtained by downsampling the input and running one LDM denoising pass, yielding coherent low-frequency structure. Finally, a lightweight fine-tuning of the VAE decoder (with skip connections) is optionally applied to alleviate compression losses and sharpen details.

**Strengths:**

1. The paper's approach appears novel. While diffusion priors have been used in low-level vision, adapting them to ultra-high-res restoration with plug-and-play guidance is new.
2. The method adopts a Plug-and-play design, which can be compatible with different diffusion models.
3. The paper is generally well-organized and clearly written.

**Weaknesses:**

1. The paper should further clarify the distinction between high-resolution image restoration and high-resolution image generation. To my knowledge, DemoFusion is a method for high-resolution generation rather than restoration. The key difference lies in that restoration tasks emphasize maintaining consistency between the low-resolution and high-resolution images, whereas training-free high-resolution generation methods do not impose such a constraint.
2. Computational cost: diffusion inference with dozens of steps and many patches is slow and resource-intensive. Indeed, patch-based diffusion on 8K images is inherently heavy.
3. FFSG is designed for U-Net-based LDMs (with self-attention). Emerging diffusion architectures (e.g. Diffusion Transformers) may not readily accommodate the same attention-based feature injection. A more important point is that the DiT architecture is currently the mainstream (such as SD3 and FLUX), which limits the application of the method presented in this paper.
4. This method introduces several new hyperparameters. The paper only provides default values for these parameters, but does not offer any sensitivity analysis on how these hyperparameters affect the final results. This makes it impossible for readers to determine whether the method is easy to adjust and whether a large amount of cumbersome parameter tuning is required when applied to new tasks or backbone models.
5. The paper lacks comparisons with the latest state-of-the-art super-resolution or image restoration models, which makes it difficult to fully assess the effectiveness of the proposed method.

**Questions:**

See Weaknesses

---

> ### Author Response · Authors · 2025-11-21
> **Response to Reviewer gDB6 (part 1/3)**
>
> Thank you for the detailed and thoughtful review. We address the reviewer’s concerns and questions point by point below.
>
> >`w1: Distinctions between high-resolution restoration and high-resolution generation`
>
> We thank the reviewer for this highly insightful comment. We fully agree that the fundamental distinction between restoration and generation lies in the constraint of consistency: restoration requires strict faithfulness to the degraded input, whereas generation prioritizes perceptual quality without such strict constraints. Based on this insight, we have summarized two core principles that distinguish our restoration-oriented design from generation methods like DemoFusion:
>
> **Principle 1: Selective Information Injection.** Generation-oriented methods like DemoFusion aim to synthesize visually coherent high-resolution content beyond the model’s native scale. In such settings, reference information is used broadly to maintain overall coherence, and deviations from the input are acceptable as long as the output remains perceptually plausible. In contrast, restoration demands selective and controlled use of reference signals. Injecting too much reference information may reintroduce degradation patterns or conflict with the true underlying structure. Our FreqG adheres to this principle by injecting only phase information to provide global structural cues without overriding the model’s ability to recover valid high-frequency details.
>
> **Principle 2: Strict Structural Fidelity.**  For generation, the main requirement is to avoid severe structural artifacts (e.g., distorted objects or repeated structures) when producing images at higher resolutions; minor variations from the conditioning image are acceptable. Restoration, however, requires the output to remain precisely faithful to the degraded input, both in structure and semantics. Any hallucinated content or unnecessary alteration directly undermines the restoration objective. Our FeatG respects this principle by injecting global contextual cues to suppress artifacts while ensuring that restored details remain consistent with the input rather than introducing new or unrelated textures.
>
> We appreciate the reviewer for highlighting this fundamental distinction, and we will refine the discussion in the final version to better articulate these principles.
>
>
>
> >`w2: Computational overhead of patch-based diffusion`
>
> We appreciate the reviewer’s observation regarding the computational cost of diffusion inference. We fully agree that multi-step sampling combined with patch-based processing makes ultra-high-resolution restoration, particularly at resolutions approaching 8K, inherently heavy. This complexity primarily reflects the characteristics of modern diffusion backbones rather than the design of our guidance mechanism.
>
> To quantify the efficiency of our approach, we provide a detailed comparison of parameters, FLOPs, peak GPU memory usage, and inference latency in **Appendix D**. The results show that FreeAdapt introduces only minimal computational overhead relative to the standard patch-based inference pipeline. For instance, LDM-Ours increases FLOPs by approximately 7% compared to LDM-PI, yet offers a substantial improvement in perceptual quality. This demonstrates that the proposed frequency–feature guidance yields a highly favorable cost–performance balance.
>
> The fundamental sources of computational cost in 8K diffusion restoration can be attributed to three intrinsic factors:  (1) the iterative sampling procedure required by diffusion models, which involves dozens of denoising steps;  (2) the large number of parameters in modern pre-trained generative backbones, which increases the computation per step; and  (3) the redundancy introduced by overlapping patches, which must be processed repeatedly to ensure seamless spatial consistency at extremely high resolutions. Within this unavoidable cost regime, FreeAdapt does not introduce additional multi-scale passes or multi-stage fusion, which enables it to remain more efficient than existing training-free diffusion adaptations while still providing superior restoration fidelity.
>
> Future work will focus on reducing the intrinsic cost of diffusion inference itself, including step distillation, architectural acceleration, and low-bit quantization, in order to further improve the practicality of UHD and 8K diffusion-based restoration in resource-constrained environments.

---

> ### Author Response · Authors · 2025-11-21
> **Response to Reviewer gDB6 (part 2/3)**
>
> >`w3: Applicability of FFSG to DiT architectures`
>
> We appreciate the reviewer’s forward-looking and highly insightful comment. We agree that DiTs are becoming the mainstream architecture, and our current implementation is built on U-Net–based LDMs. This is indeed a limitation, which we have acknowledged in Sec. 5. Nevertheless, we emphasize that the principles underlying FFSG are not inherently tied to U-Net, and the framework has a clear path toward extension to DiT architectures. Below we clarify this in two parts.
>
> FreqG is architecture-agnostic and directly applicable to DiT. FreqG is fundamentally an input-level operation that manipulates the latent ztz_tzt​ via FFT-based phase injection before the denoising step. Because this modification is applied directly to the 2D latent before tokenization, it integrates seamlessly with DiT models: the fused latent is simply fed into the standard patchification/tokenization stage. Thus, FreqG does not rely on U-Net structures and can be used with DiT without any architectural change.
>
>
> Although FeatG in its current form relies on U-Net self-attention, its core idea of combining local detail reconstruction with global reference context can be transferred to a transformer architecture by introducing a conceptual “FreeAdapt-DiT Block” that serves as a replacement for the standard DiT block：
>
> - **Input:** The block receives two token sequences: the current high-resolution patch tokens ($X_{HR}$) and the global low-resolution reference tokens ($X_{LR}$), where $X_{LR}$ is shared across patches.
> - **Step 1 (Local Branch):** Compute standard Self-Attention on $X_{HR}$ to capture local details:$$Attn_{local} = \text{SelfAttention}(X_{HR})$$
> - **Step 2 (Global Branch):** Compute Cross-Attention of $X_{HR}$ attending to $X_{LR}$ to inject global context:
> $$Attn_{global} = \text{CrossAttention}(Q=X_{HR}, K=X_{LR}, V=X_{LR})$$
> - **Step 3 (Blending):** Linearly blend the outputs of these two attention mechanisms, strictly following the logic of Eq. 9 in our paper:
> $$X_{blended} = (1-\alpha) \cdot Attn_{local} + \alpha \cdot Attn_{global}$$
> - **Step 4 (Output):** Pass $X_{blended}$ to the subsequent MLP layers of the DiT block.
>
> Recent studies (e.g., DiT4SR[1], DiTCtrl[2]) have demonstrated that DiT attention mechanisms are functionally analogous to U-Net and support similar dual-stream control strategies. We emphasize that the proposed FreeAdapt-DiT block is a conceptual extension rather than a finalized implementation. Its feasibility requires further empirical validation and potential architectural refinement, which we consider an important avenue for future work.
>
> [1] DiT4SR: Taming Diffusion Transformer for Real-World Image Super-Resolution. ICCV2025.
>
> [2] DiTCtrl: Exploring Attention Control in Multi-Modal Diffusion Transformer for Tuning-Free Multi-Prompt Longer Video Generation. CVPR2025.
>
>
> >`w4: Hyperparameter sensitivity analysis`
>
> We thank the reviewer for raising this important concern regarding the sensitivity of the introduced hyperparameters. To address this, we conducted a comprehensive sensitivity analysis, presented in **Appendix B**. Our evaluation focuses on the two key hyperparameters that govern the proposed synergistic guidance mechanism: the filter coefficient $c$ in FreqG and the fusion weight $\alpha$ in FeatG.
>
> The results demonstrate that FreeAdapt is highly robust across a wide range of values. The performance curves for both $c$ and $\alpha$ exhibit broad stable regions, indicating that the method is not sensitive to moderate changes in these parameters. In particular, the default configuration ($c = 0.15$, $\alpha = 0.20$) consistently provides strong performance across datasets, tasks, and backbone models, showing that the method generalizes well without requiring task-specific tuning.
>
> For the FreqG filter coefficient, values within $c \in [0.10, 0.20]$ produce nearly identical performance, with only extreme values leading to minor structural inconsistencies or reduced texture diversity. For the FeatG fusion weight, a similar plateau exists within $\alpha \in [0.15, 0.25]$, balancing global-context correction and local-detail fidelity. These observations confirm that FreeAdapt does not rely on finely tuned hyperparameters and remains stable under a range of upscaling settings, including high-resolution scenarios such as 8K restoration. Overall, the hyperparameter behavior of FreeAdapt is predictable, robust, and easy to configure, and the default settings already offer strong performance without cumbersome parameter tuning.

---

> ### Author Response · Authors · 2025-11-21
> **Response to Reviewer gDB6 (part 3/3)**
>
> >`w5: Comparisons with recent SR and IR models`
>
> We sincerely thank the reviewer for this important suggestion. We agree that including comparisons with state-of-the-art super-resolution and image restoration models is essential for assessing the competitiveness of our approach. In the revised version, we introduce StableSR, a representative LDM-based super-resolution method, into the UHD-IR evaluation setting, together with its extended variants (e.g., DemoFusion, MultiDiffusion, PixelSmith, and our FreeAdapt-based variant), all evaluated under the same diffusion backbones and experimental protocol as used for LDM and DiffBIR. Additionally, we report results for DreamUHD as a conventional UHD-IR baseline. The updated results can be found in **Table 1** and **Table 2**.
>
> We would also like to clarify the positioning of our work. FreeAdapt is not a standalone restoration architecture but a plug-and-play guidance framework designed to enhance pre-trained latent diffusion models by addressing the structural inconsistencies and artifact patterns that arise from patch-based sampling at ultra-high resolutions. Therefore, it is most appropriate to evaluate FreeAdapt on strong diffusion-based backbones such as LDMs and DiffBIR, and similarly on StableSR, which follows the same LDM design paradigm. As shown in **Table 2**, integrating FreeAdapt consistently improves all these diffusion models across UHD restoration tasks, demonstrating both the generality and the practical value of our framework.

---

### Official Review · Reviewer_B33R · 2025-11-09

**Soundness:** 2
**Presentation:** 3
**Contribution:** 2
**Rating:** 4
**Confidence:** 4

**Summary:**

The paper proposes FreeAdapt, a training-free framework to adapt pre-trained diffusion models (e.g., LDM, DiffBIR) to ultra-high-definition image restoration (UHD-IR) tasks. The method first generates a structure-preserving reference image at a lower resolution, then during UHD latent denoising, applies (1) frequency-domain guidance that fuses low-frequency phase information from the reference to enforce global structural and color consistency across patches, and (2) feature-level guidance that injects reference features into self-attention to provide global context and suppress hallucinated textures. In addition, an optional lightweight VAE decoder fine-tuning module with multi-scale skip connections and LoRA is introduced to better reconstruct high-frequency details. Experiments on several UHD-IR benchmarks show consistent gains over both traditional UHD restoration networks and other training-free diffusion-based adapters, especially in perceptual metrics and visual fidelity.

**Strengths:**

1. The paper proposes a novel training-free framework that adapts diffusion priors to ultra-high-resolution image restoration via designed FFSG mechanism.
2. The plug-and-play design can be seamlessly applied to different diffusion-based restoration models.
3. The paper is clearly written and well structured.

**Weaknesses:**

1. While the paper correctly reviews MultiDiffusion, DemoFusion, and PixelSmith as training-free methods for high-resolution generation, these are later treated as baselines for UHD restoration without a dedicated discussion of the additional consistency constraints that restoration requires. It would be helpful to more explicitly clarify in what sense these generation-oriented methods are adapted to restoration, and how input–output consistency is enforced or evaluated beyond standard full-reference metrics.
2. This paper lacks comparisons with the latest state-of-the-art uper-resolution or image restoration methods; many works mentioned in the related work section (e.g., DreamUHD, StableSR) are not included in the experimental evaluation, which makes it difficult to fully assess the competitiveness of the proposed approach.
3. FFSG is implemented by directly modifying the self-attention blocks in the U-Net decoder, and the paper does not provide any discussion or experiments on how to extend the proposed guidance mechanism to diffusion transformers. Given that DiT-style models are becoming the mainstream choice for strong diffusion priors, this U-Net dependency may substantially restrict the method’s impact.

**Questions:**

See Weaknesses.

---

> ### Author Response · Authors · 2025-11-21
> **Response to Reviewer B33R (part 1/2)**
>
> Thank you for the clear and detailed evaluation. We address the reviewer’s concerns and questions point by point below.
>
> >`w1: Clarify how generation-oriented diffusion methods are adapted for UHD restoration`
>
> We thank the reviewer for this insightful question. We agree that MultiDiffusion, DemoFusion, and PixelSmith were designed for high-resolution generation rather than restoration, and UHD restoration imposes much stronger consistency constraints. Below we clarify the fundamental distinction and why our frequency and feature guidance modules are specifically tailored for restoration.
>
> **Core distinction between generation and restoration.** High-resolution generation aims to synthesize visually plausible images beyond the model’s native resolution, where deviations from the input or newly introduced structures are acceptable as long as coherence is maintained. In contrast, UHD restoration must strictly preserve the geometry, layout, and semantic content of the degraded input, and any hallucinated structure contradicts the restoration objective. This essential difference explains why generation-oriented methods do not naturally enforce the structural fidelity required for UHD restoration.
>
> **Principles of restoration-oriented methods.**  A restoration-oriented framework must (1) selectively inject reliable information without overriding input-consistent details, and (2) enforce structural fidelity throughout the denoising process. These principles guide the design of our FFSG modules.
>
> Generation-oriented frequency fusion methods often mix low- and high-frequency components in ways that may unintentionally reshape structure or global appearance, which conflicts with restoration goals. FreqG provides controlled information injection by merging only the phase component from the reference while keeping the latent amplitude unchanged. Since phase encodes global structure, this operation improves structural consistency without affecting valid fine details.
>
> Restoration requires the output to remain aligned with the input at both semantic and geometric levels. FeatG is designed for this need. Rather than freely synthesizing content, it uses a dual-branch attention mechanism that blends local detail reconstruction with global contextual cues in a controlled fashion. This ensures that refinement stays faithful to the input and that global cues act mainly to suppress artifacts rather than introduce new or inconsistent textures.
>
> We will clarify these distinctions more explicitly in the revised manuscript to better distinguish generation-oriented baselines from restoration-oriented design.
>
>
> >`w2: Comparisons with recent SR and UHD-IR methods.`
>
> We thank the reviewer for raising this important point. Following the suggestion, we have now added comparisons with recent state-of-the-art UHD restoration and super-resolution models, including DreamUHD and StableSR, in all three benchmark settings. The updated quantitative results are provided in Table 1 and Table 2 (excerpted below).
>
> **Table 1:** Quantitative comparison with SOTA methods on UHD-Haze Task. (**Excerpted**)
>
> |**Method**|**PSNR ↑**|**SSIM ↑**|**LPIPS ↓**|**MUSIQ ↑**|
> |---|---|---|---|---|
> |Restormer|23.10|0.930|0.157|33.69|
> |UHDformer|22.58|0.942|0.118|31.72|
> |DreamUHD|24.36|0.945|0.116|33.08|
> |ERR|*25.10*|*0.949*|0.119|31.17|
> |StableSR-Ours|22.80|0.945|*0.092*|**43.63**|
> |DiffBIR-Ours|**25.50**|**0.953**|**0.077**|*42.18*|
>
> **Table 2:** Quantitative comparison of diffusion model adaptation methods. (**Excerpted**)
>
> |**Model**|**UHD-LL (PSNR ↑ / LPIPS ↓)**|**UHD-Haze (PSNR ↑ / LPIPS ↓)**|**UHD-Blur (PSNR ↑ / LPIPS ↓)**|
> |---|---|---|---|
> |StableSR-PI|19.14 / 0.369|19.85 / 0.182|24.10 / 0.202|
> |StableSR-Demofusion|21.59 / 0.393|20.17 / 0.211|25.31 / 0.382|
> |StableSR-Pixelsmith|20.79 / 0.361|20.92 / 0.198|25.65 / 0.297|
> |StableSR-Ours w/o VAE-FT|_21.96_ / _0.270_|_22.51_ / _0.142_|_27.07_ / _0.195_|
> |StableSR-Ours|**22.42** / **0.244**|**22.80** / **0.092**|**27.42** / **0.162**|
>
> We would also like to clarify the positioning of our method. FreeAdapt is not a standalone restoration network. Instead, it is a plug-and-play guidance framework that enhances pre-trained latent diffusion models. Its goal is to address structural inconsistencies and artifact patterns that arise from patch-based sampling when diffusion models are applied to ultra-high-resolution restoration. Therefore, the most appropriate setting to evaluate FreeAdapt is on top of diffusion-based LDM backbones, including LDMs, DiffBIR, and now StableSR (which is also LDM-based). The newly added experiments demonstrate that FreeAdapt improves all these diffusion models in a consistent manner, confirming the generality and practical value of our framework.
>
> We will update the final version of the manuscript to clearly position FreeAdapt as a diffusion-guided enhancement framework and to highlight these newly added comparisons.

---

> ### Author Response · Authors · 2025-11-21
> **Response to Reviewer B33R (part 2/2)**
>
> >`w3: Analysis on adapting FFSG to DiTs`
>
>
> We appreciate the reviewer’s insightful comment. We agree that diffusion transformers (DiTs) have become increasingly important, and our current implementation operates within the self-attention blocks of a U-Net decoder. This is a limitation we have acknowledged in Sec. 5. Below we clarify how FFSG relates to DiT architectures and why our design is not inherently restricted to U-Nets.
>
> First, the Frequency Guidance (FreqG) module is architecture-agnostic. FreqG modifies the latent $z_t$ at the input level through FFT-based phase injection. This operation takes place entirely in the 2D latent space before denoising, and therefore before the tokenization step in DiTs. As a result, FreqG can be applied to DiT models directly, without any change to the transformer layers. The fused latent is simply fed into the standard DiT pipeline.
>
> Second, the core idea of Feature Guidance (FeatG) has a natural extension to DiTs. While FeatG in our current implementation interacts with U-Net self-attention, its underlying logic is model-agnostic: combining a local attention pathway with a global reference pathway and blending their outputs. This same logic can be instantiated within a transformer block. A conceptual FreeAdapt-DiT block operates as follows:
>
> The local branch computes the standard self-attention on the high-resolution tokens $X_{HR}$:
>
> $$Attn_{local} = \text{SelfAttention}(X_{HR}).$$
> For the global branch, the high-resolution tokens attend to the reference keys and values derived from the native-resolution reference latent:
>
> $$Attn_{global} = \text{CrossAttention}(Q=X_{HR}, K=X_{LR}, V=X_{LR}).$$
>
> The two attention outputs are finally combined through a linear blending scheme:
>
> $$Attn_{final} = (1-\alpha) \cdot Attn_{local} + \alpha \cdot Attn_{global},$$
>
> which preserves the local detail modeling while incorporating global semantic cues. This formulation aligns with the functional behavior of FeatG and can be mapped onto transformer blocks without altering the fundamental principles of our guidance mechanism. Existing studies such as DiT4SR[1] and DiTCtrl[2] demonstrate that transformer-based diffusion models support attention architectures that incorporate multiple information streams, suggesting that this type of extension is structurally feasible.
>
> We emphasize that the transformer adaptation described above is a conceptual pathway. Although the guiding principles of FFSG extend naturally beyond U-Net architectures, developing and validating a complete DiT implementation will require further architectural investigation. We view this as an important and meaningful direction for our future work.
>
> [1] DiT4SR: Taming Diffusion Transformer for Real-World Image Super-Resolution. ICCV2025.
>
> [2] DiTCtrl: Exploring Attention Control in Multi-Modal Diffusion Transformer for Tuning-Free Multi-Prompt Longer Video Generation. CVPR2025.

---

### Author Response · Authors · 2025-11-21
**Summary of Changes**

We appreciate all valuable feedback and comments from reviewers. The revised manuscript includes the following key updates, with all modifications marked **in blue** for clarity:
1. **Section 2:** Added a clear explanation of the fundamental differences between high-resolution image generation and high-resolution image restoration, emphasizing the consistency constraints unique to restoration.
2. **Section 3:** Revised the descriptions of the reference-image generation pipeline and the VAE-FT module to provide a clearer, more concise explanation of their roles within FreeAdapt.
3. **Experimental Evaluation:**
    - Added comparisons with DreamUHD and StableSR to complete the evaluation against recent UHD-IR and diffusion-based baselines (Table 1 and Table 2).
    - Summarized the performance gains contributed by FreeAdapt across multiple LDM-based restoration backbones (Table 2).
    - Added a UHD image deraining experiment on the 4K-Rain13k dataset, including quantitative results (Table 6) and qualitative visualizations (Figure 9).
4. **Additional Analyses:**
    - Added hyperparameter sensitivity analysis for the main FFSG components (Figure 8).
    - Added computational efficiency statistics, including VRAM usage and runtime measurements (Table 5).
5. **Discussion & Applicability:**
    - Added  application scenarios where quality-centric UHD restoration is practical.
    - Added a discussion on extending FFSG to diffusion transformers (DiTs) and provided a conceptual pathway for adaptation.

These updates enhance the clarity, completeness, and practical relevance of the revised submission. We sincerely thank the reviewers again for their constructive comments.

---

### Meta-Review · Area_Chair_qP95 · 2026-01-02

**Summary:**

This submission presents FreeAdapt, a plug-and-play framework that leverages diffusion priors for Ultra-High-Definition Image Restoration  through a training-free Frequency-Feature Synergistic Guidance mechanism. The four reviewers provided initial scores of 4, 4, 6, and 4, with Reviewer PoKA subsequently raising their score to 6 after the rebuttal. Key concerns raised by reviewers included: insufficient comparisons with recent state-of-the-art UHD-IR and super-resolution methods, confusion about the "training-free" claim due to the optional VAE fine-tuning module, lack of computational efficiency analysis and hyperparameter sensitivity studies, missing UHD deraining experiments,  unclear distinction between restoration and generation tasks, questions about applicability to Diffusion Transformer architectures , and concerns about similarity to FAM Diffusion.

**Reviewer Concerns:**

The authors' rebuttal added  new experiments including comparisons with DreamUHD and StableSR, UHD deraining experiments on the 4K-Rain13k dataset, detailed computational cost analysis, and systematic hyperparameter sensitivity studies. The authors clarified the distinction between restoration and generation, explaining that restoration requires strict input-output fidelity while generation methods prioritize perceptual plausibility. They also provided a conceptual pathway for extending FFSG to DiT architectures and clearly differentiated their method from FAM Diffusion in terms of frequency injection strategy, attention formulation, resolution handling, and task objectives.

The primary concern that remains partially outstanding relates to evaluation on real-world or AI-generated benchmarks (raised by Reviewer H5xg). The authors acknowledged the scarcity of paired UHD datasets and identified this as an important direction for future work. Regarding the "training-free" claim, the authors clarified through ablation studies that FFSG provides the primary performance gain, while the optional VAE-FT module offers only secondary refinement, and that VAE-FT operates at the task level rather than model-specific adaptation.

**Reviewer Scores:**

The overall trajectory of the discussion, particularly Reviewer PoKA's explicit score increase and the comprehensive nature of the authors' responses, supports an accept decision for this technically sound  contribution to UHD image restoration using diffusion priors.

---

### Decision · Program_Chairs · 2026-01-26

Accept (Poster)